# Deep learning of human polyadenylation sites at nucleotide resolution reveals molecular determinants of site usage and relevance in disease

Emily Kunce Stroup[1] & Zhe Ji [1,2] ✉

The genomic distribution of cleavage and polyadenylation (polyA) sites should be co-evolutionally optimized with the local gene structure. Otherwise, spurious polyadenylation can cause premature transcription termination and generate aberrant proteins. To obtain mechanistic insights into polyA site optimization across the human genome, we develop deep/machine learning models to identify genome-wide putative polyA sites at unprecedented nucleotide-level resolution and calculate their strength and usage in the genomic context. Our models quantitatively measure position-specific motif importance and their crosstalk in polyA site formation and cleavage heterogeneity. The intronic site expression is governed by the surrounding splicing landscape. The usage of alternative polyA sites in terminal exons is modulated by their relative locations and distance to downstream genes. Finally, we apply our models to reveal thousands of disease- and trait-associated genetic variants altering polyadenylation activity. Altogether, our models represent a valuable resource to dissect molecular mechanisms mediating genome-wide polyA site expression and characterize their functional roles in human diseases.

Cleavage and polyadenylation is an essential step of mRNA maturation and comprises two steps: endonucleolytic cleavage of a nascent transcript followed by polyA tail synthesis. The process is crucial for transcription termination, and the underlying regulatory mechanisms are tightly coupled with gene transcription and RNA splicing[1]. One gene can contain multiple polyA sites leading to alternative polyadenylation (APA) events that generate RNA isoforms with different 3' untranslated regions (3'UTRs) or coding sequences. Transcript isoforms with the same coding sequence but different 3'UTRs can show variable translation efficiency, stability, and subcellular localization. APA plays important regulatory roles in gene expression during various physiological processes, such as oncogenesis, development, and stress response[2–7].

The mammalian polyadenylation complex is composed of ~20 core and 60 auxiliary proteins[8,9]. These RNA-binding proteins form subcomplexes that recognize unique *cis*-regulatory elements (Supplementary Fig. 1a)[10]. The polyadenylation signal (PAS; AAUAAA or variants), located 10-30 nucleotides (nt) upstream of the cleavage site, is recognized by the cleavage and polyadenylation specificity factor (CPSF) complex. The U-rich and GU/CU-rich elements, located downstream of the cleavage site, are bound by the cleavage stimulation factor (CstF) dimer. Several other motifs play regulatory roles, including UGUA (bound by cleavage factor I (CFI)) and U-rich elements (recognized by FIP1) around the PAS, as well as G-rich elements (bound by hnRNPh/f) downstream of CstF binding. Although these motifs are known to promote polyA site formation, there is still a lack of

[1]Department of Pharmacology, Feinberg School of Medicine, Northwestern University, Chicago, IL 60611, USA. [2]Department of Biomedical Engineering, McCormick School of Engineering, Northwestern University, Evanston, IL 60628, USA. ✉e-mail: zhe.ji@northwestern.edu

quantitative measurements of their position-specific importance and the crosstalk among the motifs. Building a quantitative model would be valuable to study the physiological roles of polyadenylation because genetic variants altering these motifs can regulate human gene expression and disease progression. For example, mutations of the PAS AAUAAA in genes such as *TP53* (associated with cancer susceptibility), *HBA2* (linked to α-thalassemia), and *INS* (associated with neonatal diabetes) can cause the downregulation of protein production and lead to disease initiation[11–14].

Besides the motif strength, polyadenylation activity is tightly regulated by the local gene structure. During genome evolution, the localization of putative polyA sites must be optimized within the genomic context. Due to the competition with splicing, intronic polyadenylation happens when the splicing activity is weak (e.g., weak 5' splice sites (5'SSs))[15,16]. Efficient polyadenylation is essential for proper transcription termination especially when the tandem intergene distance is short[17]. Otherwise, the run-on of RNA polymerase II can interfere with the downstream gene expression. Quantitative modeling of polyA site expression which considers the genomic context would provide molecular insights into site optimization across the genome.

In this study, we aimed to develop a computational approach to unbiasedly identify genome-wide putative polyA sites and then examine the local genomic parameters determining their expression. The deep learning approach is well suited to studying the motif "grammar" underlying polyadenylation. Convolutional neural networks can quantitatively capture the dynamic interactions among *cis*-regulatory motifs. Although previous studies have developed several models to predict polyA site occurrence and strength, these models have their limitations. For example, the DeepPASTA model[18] calculates the probability of a sequence being a polyA site, but it does not perform nucleotide-level cleavage site prediction. The APARENT[19,20] and PolyApredictors[21] models predict the polyA site strength and cleavage profile, but they were restricted to database-annotated polyA sites and anchored with the predefined PAS (e.g. AAUAAA/AUUAAA) for the prediction. To address these technical limitations, here we developed the first deep learning model which performs unbiased identification of putative polyA sites at nucleotide resolution across the human genome and a separate model to calculate the site strength. The models quantitatively revealed the crosstalk among *cis*-regulatory motifs in determining polyA site formation and cleavage heterogeneity. Furthermore, using logistic regression, we examined genomic parameters determining polyA site expression in introns and terminal exons. Finally, we demonstrated that our models can be used to characterize disease/trait-associated genetic variants regulating polyadenylation activity.

## Results

### Building a deep learning model named PolyaID to identify polyA sites at nucleotide resolution

To obtain expressed polyA sites for the model training, we analyzed a large cohort of publicly available 3' Region Extraction and Deep Sequencing (3'READS) data[22,23]. We chose this method because it resolved the internal priming issue and its polyA site supporting (PASS) reads reliably quantify genome-wide polyA site expression at nucleotide resolution (Fig. 1a, see "Methods" for details). In total, we analyzed 1.5 billion sequencing reads from 103 human samples across 19 different tissue/cell types and selected 299 million PASS reads to map the polyA sites (Supplementary Data 1). Out of 109,814 polyA sites identified, 44.2% used the canonical PAS AAUAAA, 14.9% had AUUAAA, and the remaining 40.9% used other PAS types (Supplementary Fig. 1b-c, Supplementary Data 2).

For the PolyaID model training, we used the 240 nt sequences around the 3'READS-identified polyA sites as positive examples, and we included the sequences from each position from −25 to +25 nt

surrounding the maximum cleavage site to augment the training dataset. Our negative examples included random genomic sequences, shuffled transcript sequences, and neighboring sequences 50–100 nt distant from polyA sites that did not contain any PASS reads in the middle 50 nt regions. Different from previously published models, we included many site-adjacent sequences in the positive (<25 nt from the cleavage site) and negative (50–100 nt away from the site) sets, respectively. This design was important for our algorithm to robustly identify polyA sequences showing cleavage in the middle region. And the model is not biased to any predefined PAS types.

The sequence modeling unit of our PolyaID model contained one convolutional layer and one bidirectional long short-term memory (LSTM) layer (Fig. 1b). The model has two output branches: one outputs the classification probability that a sequence represents a polyA site; the other predicts the cleavage probability vector (50 nt long) surrounding the center position (Fig. 1b). The 3'READS-measured cleavage probabilities within the 50 nt region around polyA sites were used for model training. Detailed model parameters are described in Supplementary Data 3. This design allowed us to model the nucleotide-resolution cleavage probability coupled to site identification, accounting for the imprecision inherent to the cleavage and polyadenylation processes[22].

Our PolyaID model achieved an area under the receiver operating characteristic curve (AUROC) of ~0.974 on the split training, validation, and testing datasets (Fig. 1c and Supplementary Fig. 2a). The area under the precision-recall curve (AUPRC) was ~0.98 for all splits (Supplementary Fig. 2a). PolyaID performed well to identify sites expressed at different levels (Fig. 1d). Compared to published deep learning models (i.e. DeepPASTA[18], APARENT[19], APARENT2[20], and PolyApredictors[21]), PolyaID showed higher AUROC values in classifying polyA sequences with different PAS signals vs. random nucleotides (i.e. AAUAAA, AUUAAA, or others) (Supplementary Fig. 2b–c). Especially, our PolyaID model is not restricted to predictions using sequences centered at the maximum cleavage site or PAS. It outperformed other algorithms when we used input sequences randomly shifted (−25 to +25 nt) from the maximum cleavage site (ΔAUROC > 0.1; Supplementary Fig. 2d–e). The trend is consistent when we evaluated model performance in classifying defined polyA sites from published databases (i.e. PolyA_DB[24] and APADB[25]) vs. random nucleotide sequences (Supplementary Fig. 2g–j). As detailed in later paragraphs, this unique advantage allowed our model to unbiasedly scan the genomic sequence and identify putative polyA sites.

We then assessed the quality of the cleavage vector predictions by calculating the mean cleavage position, which accounts for the shape of the cleavage probability vector and indicates the most likely cleavage site. In the holdout testing set, our model achieved a correlation between observed 3'READS cleavage and predicted positions of 0.963 (Fig. 1e), which is comparable to the APARENT models and is better than the PolyApredictors model (Supplementary Fig. 2f–h). Previous studies have shown that cleavage events tend to happen after the CA-dinucleotide or in genomic A-rich regions[26]. Our PolyaID predicts the cleavage probability at genomic As while APARENT did not allow cleavage happening at As (Supplementary Fig. 2k–l) (see "Methods" for details).

Next, we consolidated the classification and cleavage probability predictions from PolyaID to achieve nucleotide-resolution identification of the polyA sites across genomic regions. We used a scanning window moving across the region of interest 1 nt per step and performed a PolyaID prediction at each position using the 240 nt input nascent RNA sequence. We retained only the positive predictions that had a classification probability >0.5 and a middle cleavage probability >0.05. We then averaged the overlapping cleavage profiles based on these retained positive predictions. The position showing the maximum cleavage probability was used as the representative polyA site. This process is illustrated by two example genes: the solo polyA site of

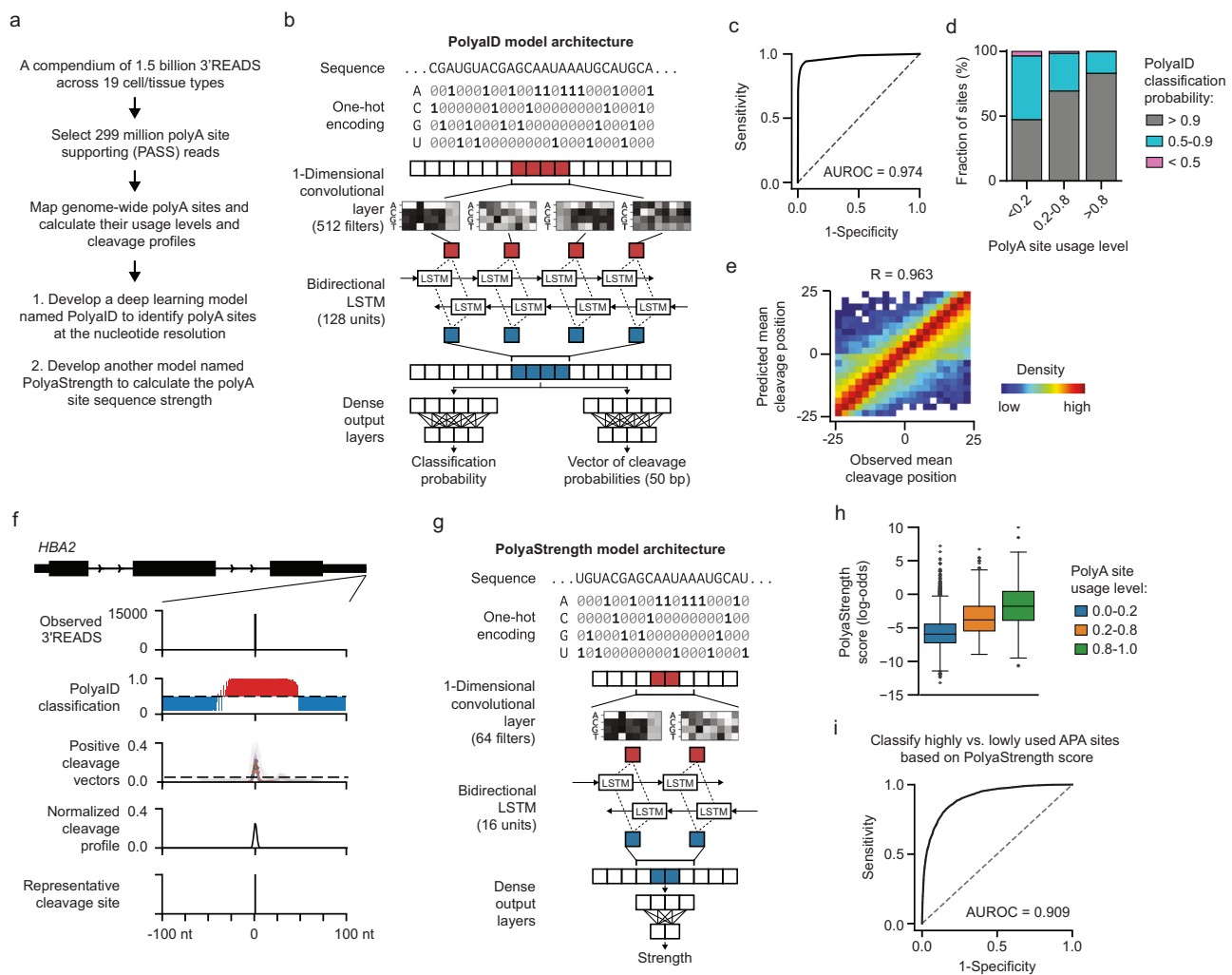

**Fig. 1 | Developing deep learning models to identify polyA sites at nucleotide-level resolution and calculate polyA site strength. a** Overview of data analysis steps. **b** The PolyaID model architecture. **c** The ROC curve shows the classification performance of the PolyaID model on the holdout test data set. The AUROC value is shown. **d** The PolyaID-predicted classification probability for terminal exonic polyA sites expressing at different levels: usage <0.2 ($N = 3886$), 0.2–0.8 ($N = 887$), usage > 0.8 ($N = 504$). See the "Methods" for the details to calculate polyA site usage levels. **e** The heatmap shows the correlation between the observed and predicted mean cleavage positions in the polyA sites of the test data set which were true positives with a PolyaID > 0.5 ($N = 91,297$). The Spearman correlation coefficient is shown. **f** The example gene *HBA2* shows our analysis procedure using the PolyaID

classification and cleavage probabilities to identify polyA sites at single-nucleotide resolution. We performed a prediction using the 240 nt nascent RNA sequence as the input with 1 nt per step. The example results between −100 to +100 nt around the polyA site were shown. **g** The PolyaStrength model architecture. **h** The correlation between the PolyaStrength score and polyA site usage level: usage <0.2 ($N = 3886$), 0.2–0.8 ($N = 887$), usage > 0.8 ($N = 504$). The boxes are bounded by the 25 and 75 percentiles and the center represents the median. The whiskers extend from each edge of the box to indicate the 1.5× interquartile range (IQR). **i** The ROC curve showing the performance using PolyaStrength score to classify highly vs. lowly expressed terminal exonic polyA sites ($N = 8167$ pairs of polyA sites).

the *HBA2* gene (Fig. 1f) and multiple polyA sites in the *GPATCH11* gene (Supplementary Fig. 2m). While the PolyaID classification probability predicted a broad positive region containing a polyA site, its combination with the cleavage probability distribution revealed the polyA site location at nucleotide resolution.

### Developing a deep learning model named PolyaStrength to calculate polyA site strength

For APA genes, one major determinant of site usage is the sequence motif strength. While our PolyaID model distinguishes polyA sites from negative sequences, it does not have a good dynamic range to separate highly vs. lowly used sites. To address this, we developed a deep learning model named PolyaStrength to calculate the polyA site strength based on the surrounding sequence (Fig. 1g). We used the relative usage levels of APA sites located in 3′ terminal exons as the proxy measurement for site strength (see "Methods" for detail). We reasoned that although one gene can contain multiple strong or weak

sites, their relative usage should capture the overall sequence strength when considering the thousands of genes expressed and using merged reads across cell/tissue types to calculate site expression.

Our PolyaStrength model used a similar sequence modeling unit as PolyaID with a single output branch predicting the site strength measured by the log-odds of polyA site usage levels (detailed parameters were described in Supplementary Data 3). The PolyaStrength scores obtained from our model were correlated with terminal exonic polyA site usage (Fig. 1h). These scores can effectively classify high vs. low usage terminal exonic sites with an AUROC of 0.909 (Fig. 1i). The performance is better than published models (Supplementary Fig. 3a). Similar results were obtained when we performed similar analyses using the polyA sites and their expression levels from PolyA_DB and APADB (Supplementary Fig. 3b, c). Moreover, we analyzed the published massively parallel reporter assay (MPRA) data[19], which introduced hundreds of thousands of randomly mutated polyA sequences to cells and quantified the site expression levels using deep

sequencing. This dataset was used to train the APARENT model. Our PolyaStrength achieved nearly comparable AUROC and AUPRC values classifying the highly vs. lowly expressed sites compared to APARENT and APARENT2, and is better than PolyApredictors (Supplementary Fig. 3d–f). The data indicated that although our PolyaStrengh model was trained based on the 3′READS data, it can be robustly applied to classify strong vs. weak sites measured by diverse methods.

## Our models revealed position-specific motif crosstalk to the polyA site definition

The convolutional network of our deep learning models extracts predictive sequence motifs and can be used to generate quantitative measurements of motif importance and the effects of motif interactions on the predictions. To obtain the positional importance of motifs, we replaced each hexamer in the 240 nt surrounding a polyA site with random hexamers and calculated the log-odds change (Δlog-odds) in the PolyaID-predicted classification probability and the PolyaStrength score. To identify hexamers contributing significantly to the genome-wide polyA sites, we calculated the importance score of a hexamer in a location by summing their −(Δlog-odds) values (Supplementary Fig. 4a). A higher importance score indicates that the motif makes a stronger contribution to genome-wide polyA site formation. The scores learned from PolyaID and PolyaStrength models were highly correlated (Pearson correlation coefficient = 0.996; Supplementary Fig. 4b). The data indicated that similar *cis*-regulatory motifs drive both polyA site formation and strength.

To characterize the motifs regulating polyadenylation activity, we used a 40 nt sliding window and identified the hexamers showing higher importance scores than expected (Supplementary Fig. 4b, see "Methods" for details). Our analyses revealed the positional importance of known *cis*-regulatory elements, such as the AAUAAA PAS and variants, UGUA, U-rich, GU/CU-rich, and G-rich motifs (Fig. 2a, b, Supplementary Fig. 4c–d, and Supplementary Data 4). AAUAAA and other PAS variants showed maximum per-site importance at 21 nt upstream of the cleavage site (Supplementary Fig. 4e). To quantify the strength differences among the PAS types, we systematically mutated AAUAAA and calculated the change in PolyaStrength scores. Mutating AAUAAA to AUUAAA caused a threefold decrease in the strength scores, whereas mutation to other PAS variants resulted in a 4.5- to 13-fold decrease (Supplementary Fig. 4f). AUAAAA, AAAUAA, and AGUAAA were the strongest among these variants (Supplementary Fig. 4f).

Upstream of the cleavage sites, the relative distance between a UGUA motif and the AAUAAA signal sequence regulates the interaction efficiency between the CFI and CPSF complexes. We examined the UGUA motif importance as a function of the distance to the AAUAAA signal (Fig. 2c). UGUAs located upstream of AAUAAA showed 2.0-fold higher importance than those located downstream. The peak occurred when the UGUA was 10–20 nt upstream (Fig. 2c). These results are consistent with a published study using biochemical assays measuring the position-dependent interactions between UGUA and AAUAAA[27]. Similarly, we examined the interactions between AAUAAA and the U-rich elements located upstream of the cleavage sites, which are bound by FIP1. The U-rich elements are 2.1-fold stronger when located downstream of AAUAAA compared with those located upstream, and the optimal distance between the two motifs was 1–6 nt (Fig. 2d).

Downstream of the cleavage sites, GU/CU- and U-rich motifs are the binding sites of the CstF dimer. The GU/CU-rich elements showed the maximum importance when they were located 10 nt downstream of the cleavage site and 24 nt from the AAUAAA (Fig. 2b, e). The U-rich elements were optimal when located immediately downstream (<8 nt) of the GU/CU-rich elements (Fig. 2f, g). G-rich elements far downstream from cleavage sites promote polyadenylation activity through the binding of hnRNPh/f[28,29]. These elements showed maximum per-

site importance when they were located 10–25 nt downstream of GU/CU-rich elements (Fig. 2h, i).

Our above analyses revealed the optimal positions of six *cis*-regulatory motifs for the binding of cleavage and polyadenylation factors (i.e. CPSF, CstF dimer, CFI, FIP1, and hnRNP) (Fig. 2j). Next, we examined whether individual genomic sites tend to use all or subset motifs. We analyzed 13,876 highly expressed polyA sites (≥100 PASS reads and usage level ≥5%) with the PAS AAUAAA (CPSF binding) and at least one downstream U-rich or GU/CU-rich element (CstF binding). The motif compositions of these polyA sites tend to be diverse (Supplementary Fig. 5a–c). Besides the CPSF and one CstF motif, 44.9% of these sites contained one additional optimally positioned motif, 17.4% of sites contain two, 2.5% of sites have three, and very few sites (0.14%) have all other four motifs (Supplementary Fig. 5a–c). Sites with more optimized motifs are generally stronger and are likely to be the single or distal sites of the genes (Supplementary Fig. 5d, e). But genes showing different polyA site motif compositions are expressed at comparable levels (Supplementary Fig. 5f). Indeed, besides the polyadenylation activity, the gene expression levels are also determined by other regulatory layers such as transcription initiation. The gene ontology analyses of genes with ≥5 optimized motifs showed that they are enriched in the pathways such as "positive regulation of macromolecule metabolic process", "cellular response to stress", and "positive regulation of transcription, DNA-templated" (Supplementary Fig. 5g). Altogether, we used our deep learning models to reveal the crosstalk among the *cis*-regulatory motifs promoting polyA site formation, and showed that genome-wide polyA sites tend to have diverse motif configurations which potentially allow differential regulation of site usage during biological processes.

## PolyaID revealed molecular mechanisms controlling cleavage site heterogeneity

The cleavage profiles of different polyA sites can be quite variable. For some sites, the cleavage events are heterogeneous and span a region >15 nt, while for others, the cleavage site is quite precise <4 nt (Fig. 3a). Currently, the molecular mechanisms regulating cleavage heterogeneity remain mostly uncharacterized. Here, we aimed to address this question using our PolyaID model because it quantifies the cleavage probabilities for a 50 nt window surrounding the maximum probability cleavage site. We developed an entropy value to measure the cleavage heterogeneity of a polyA site, and a higher entropy value indicates more heterogeneous cleavage (Fig. 3a; see "Methods" for details). To ensure reliable quantification, we only included highly expressed sites with ≥100 PASS reads in the analyses. The entropy values calculated based on our PolyaID cleavage vectors were correlated with those from PASS reads (Fig. 3b).

We next compared polyA sites with high vs. low entropy values. The polyA sites showing higher heterogeneous cleavage tend to have multiple AAUAAA sequences. 12% of high entropy polyA sites contained ≥2 AAUAAA, while only 4.5% of low entropy sites did (Fig. 3c). We did not observe significant differences in the number of other regulatory motifs (e.g., U-rich and GU/CU-rich elements) (Supplementary Fig. 6a–b). The data indicated that multiple sites for CPSF binding can lead to heterogeneous cleavage.

But the vast majority of the polyA sites (83.1% of the low entropy group and 77.6% of the high entropy group) contain only one AAUAAA. Next, we examined the differences between these sites. The high and low entropy groups showed comparable PolyaID classification probabilities (Fig. 3d), but the high entropy sites showed lower PolyaStrength scores (Fig. 3e). We divided the polyA site sequences into 4 subregions (i.e., −120/−31, −30/0, 0/+30, and +31/+120), and then examined *cis*-element occurrence differences in these subregions, comparing high vs. low entropy groups. For polyA sites showing high heterogeneous cleavage, the AAUAAA PAS and downstream U-rich elements were more likely to be located far away from the cleavage site

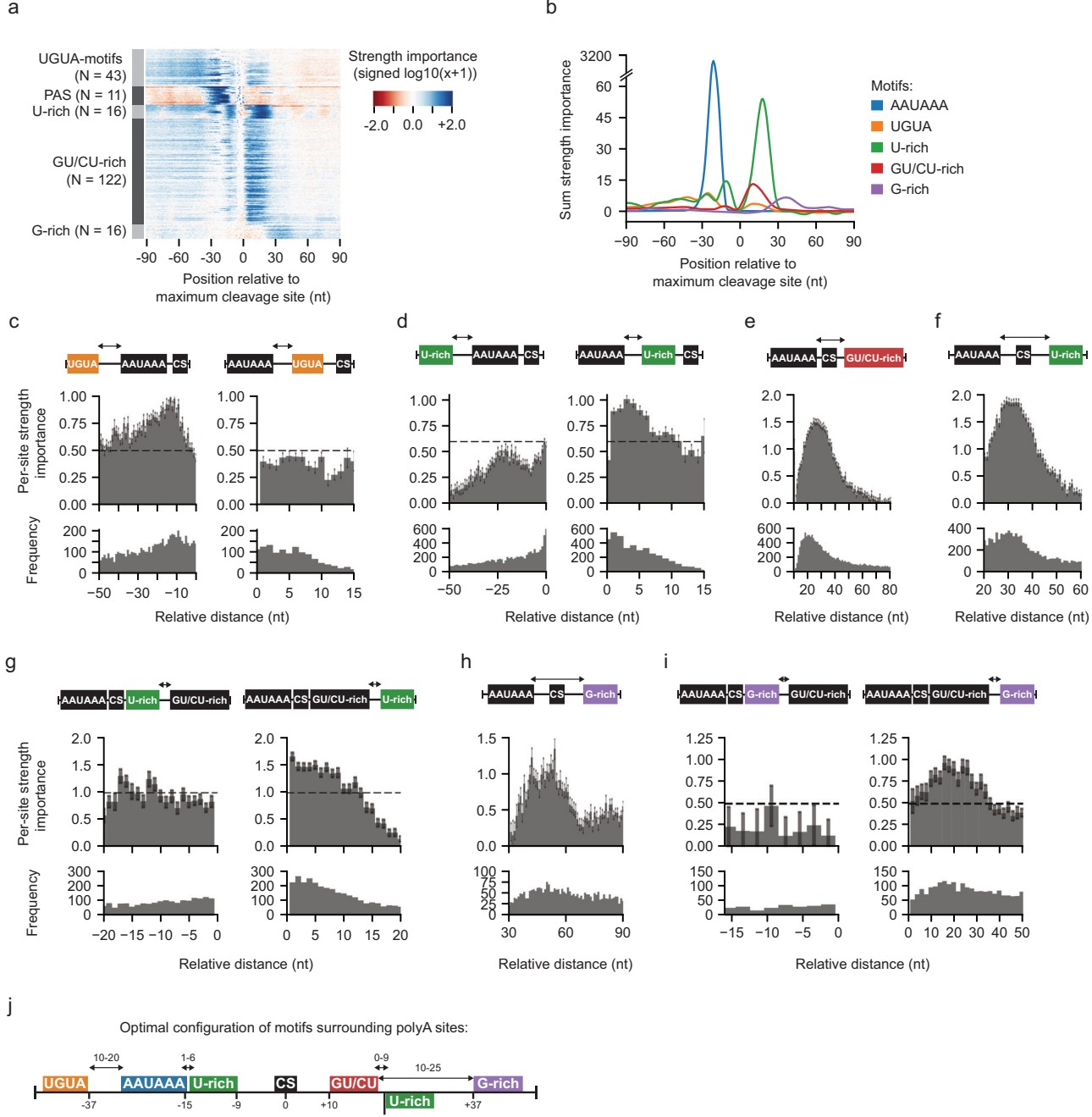

**Fig. 2 | Quantifying positional crosstalk among *cis*-regulatory elements in polyA site definition. a** The heatmap shows the strength importance (signed log10(|sum strength importance|+ 1)) for *cis*-regulatory motifs (see Supplementary Fig. 4a and "Methods" for the calculation of motif importance score). Each row represents a hexamer motif. We grouped the 222 hexamers into families as they were recognized by different cleavage and polyadenylation factors. **b** Distribution of summed importance scores per hexamer for indicated *cis*-regulatory element types. **c** We grouped the UGUA motifs based on their relative positions to the PAS AAUAAA and calculated the per-site importance score (*N* = 6141 upstream and 1273 downstream UGUA). UGUA motifs showed higher importance scores when they were located upstream of AAUAAA with a peak at −13 nt. The mean and standard error bars of per-site importance scores, as well as the motif frequency, are shown. **d** Similar to (**c**), we examined the importance score of U-rich elements located upstream of cleavage sites (*N* = 9675 upstream and 4080 downstream U-rich

elements). **e** The importance score of GU/CU-rich elements located downstream of the cleavage sites, grouped based on their relative distance from the AAUAAA signal sequence (*N* = 15,011). The mean and standard error bars of per-site importance scores, as well as the motif frequency, are shown. **f** Similar to (**e**), we examined the per-site importance score of U-rich elements (*N* = 8892). **g** Similar to (**c**), we studied positional localization effects U-rich and GU/CU-rich elements downstream of the cleavage sites (*N* = 1868 U-rich elements before and 3015 U-rich elements after GU/CU-rich motifs). **h** Similar to (**e**), the relative distances between AAUAAA and G-rich elements were examined (*N* = 2991). **i** Similar to (**c**), we studied positional localization effects G-rich and GU/CU-rich elements downstream of the cleavage sites (*N* = 244 G-rich elements before and 2201 G-rich elements after GU/CU-rich motifs). **j** Schematic showing the optimal configuration of motifs within polyA sites based on the above analyses.

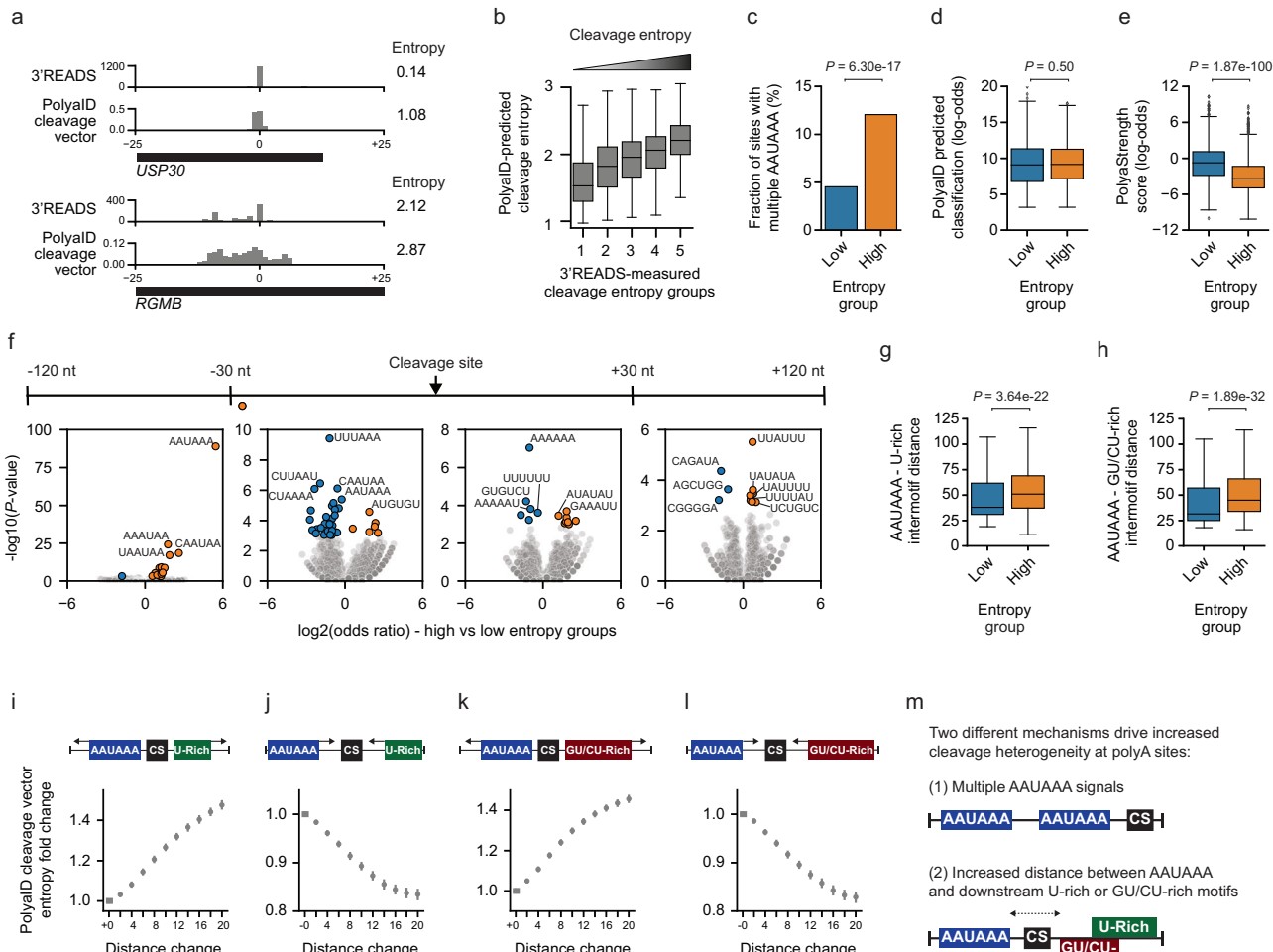

**Fig. 3 | Examining the molecular determinants of cleavage site heterogeneity.**
**a** Examples of the 3'READS distribution and normalized predicted cleavage vector for a low entropy site (*USP30*, top) and high entropy site (*RGMB*, bottom). Calculated entropy values are shown. **b** The correlation between observed entropy values calculated using 3'READS and the entropy scores calculated based on PolyaID-predicted cleavage profile (*N* = 4226 for each group). The boxes are bounded by the 25 and 75 percentiles and the center represents the median. The whiskers extend from each edge of the box to indicate the 1.5× interquartile range (IQR). **c** Fraction of sites in the low (blue, *N* = 2031) and high (orange, *N* = 1783) entropy groups that contain multiple AAUAAA PAS. The *P*-value of the two-sided hypothesis test comparing the two proportions is shown. **d** Log-odds PolyaID-predicted classification probability comparing entropy groups (*N* = 2031 low entropy sites and 1383 high entropy sites with a single AAUAAA signal). See (**b**) for boxpot definition. The two-sided Wilcoxon rank sum test *P*-value is shown. **e** Similar to (**d**), showing log-odds PolyaStrength scores. **f** Hexamer enrichment in different polyA site regions comparing low vs. high entropy groups. The polyA sites with a

single AAUAAA were used (*N* = 1687 low and *N* = 1383 high entropy). The *y*-axis shows the -log10(two-sided Chi-squared test *P*-value), and the *x*-axis indicates the log2(odds ratio values). Motifs are highlighted in orange if enriched in high entropy sites or blue if enriched in low entropy. **g** The AAUAAA − U-rich motif distance for low and high entropy groups (*N* = 1178 and 1030, respectively). Distance is measured from the last upstream AAUAAA to the first U-rich motif downstream. See (**b**) for boxpot definition. The two-sided Wilcoxon rank sum test *P*-value is shown. **h** Similar to (**g**), we show the AAUAAA−GU/CU-rich distance for low and high entropy groups (*N* = 1528 and 1246, respectively). See (**b**) for boxpot definition. **i−l** We moved the AAUAAA and U-rich (or GU/CU-rich) elements farther apart or closer together (1 nt each per step), and then examined the fold change of entropy values calculated by the PolyaID model (*N* = 676 AAUAAA-U-rich motif pairs moved farther apart (**i**), 635 moved closer (**j**), 949 AAUAAA-GU/CU-rich motif pairs moved farther apart (**k**), and 658 moved closer (**l**)). The mean and 95% confidence interval of the fold change values are shown. (**m**) The two mechanisms regulating cleavage heterogeneity.

(>30 nt) (Fig. 3f and Supplementary Fig. 6c−e). In addition, the distances between AAUAAA and downstream GU/CU- and U-rich elements were significantly longer in high vs. low entropy groups (Wilcoxon Rank-sum test $P < 10^{-21}$, Fig. 3g, h). These data indicate that overly distant binding of the CPSF and CstF complexes can lead to heterogeneous cleavage between them.

We recapitulated this regulation using our PolyaID model. When we moved the PAS and downstream U-rich and GU/CU-rich elements closer, the cleavage became more precise as indicated by decreased entropy scores, and vice versa (Fig. 3i−l and Supplementary Fig. 6f−i). To further validate the results, we reanalyzed the MPRA data[19] and found that polyA sites containing multiple AAUAAA and those with longer distances between AAUAAA and downstream U-rich or GU/CU-rich elements showed higher cleavage heterogeneity (Supplementary

Fig. 6j−l). Taken together, we showed that the number of PASs and the distance between CPSF and CstF binding sites can both regulate the cleavage heterogeneity of polyA sites (Fig. 3m).

## The differential expression of putative polyA sites in sub-gene regions

Using the PolyaID model, we comprehensively identified putative polyA sites across RefSeq-defined gene regions and 5 kb 3'UTR-extended regions using the nascent RNA sequences as the input for the prediction. In total, we identified 3.6 million putative sites with PolyaID-predicted classification probability >0.5 in 32,014 genes (Fig. 4a and Supplementary Fig. 7a−b). On average, there were 112 putative polyA sites per gene and 2 sites per kb of genomic sequence (Supplementary Fig. 7c). Out of these, only 3.0% of the sites were

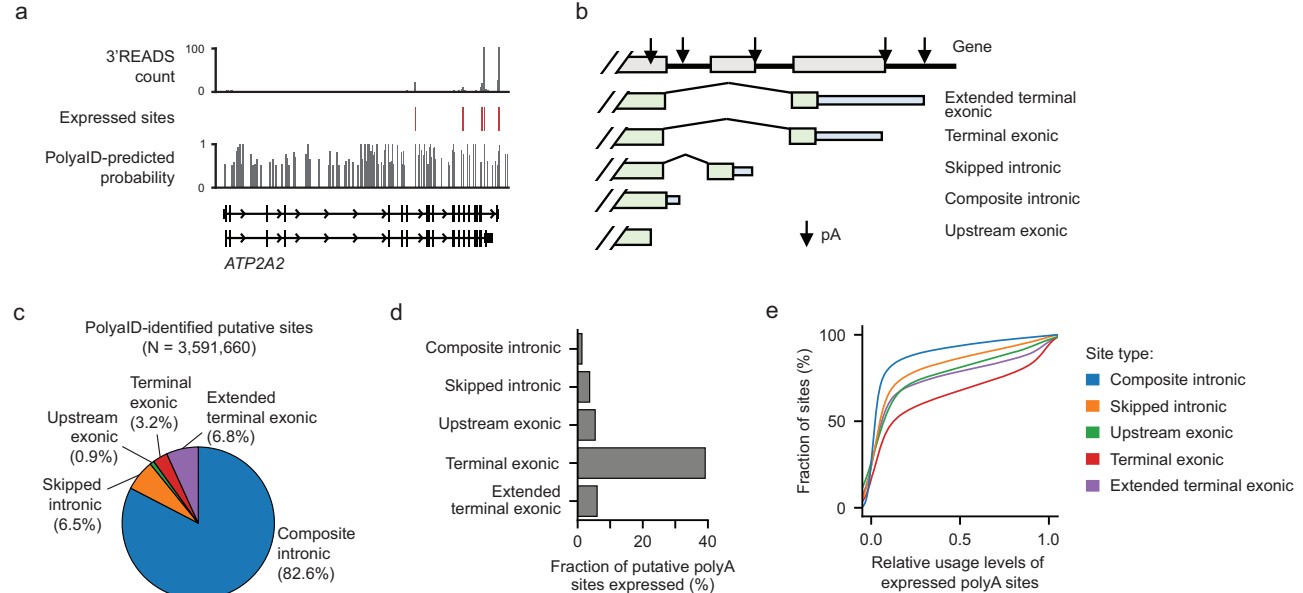

**Fig. 4 | The distribution and expression of genome-wide putative polyA sites identified by PolyaID. a** An example gene, *ATP2A2*, shows the location of PolyaID-predicted putative sites, 3'READS distribution, and the expressed sites supported with ≥10 PASS reads. **b** Schema showing different types of polyA sites defined based on gene structures. **c** Genomic distribution of PolyaID-identified putative polyA sites (classification probability >0.5). The total number and fraction of sites located in different regions are shown. **d** The fraction of putative polyA sites expressed (supported by ≥10 PASS reads). **e** The CDF plot of expressed polyA site usage levels. PolyA sites were grouped based on their genomic locations. We only considered genes with ≥2 polyA sites. The relative usage of a polyA site was calculated as the ratio between the number of reads supporting the site and the summed reads from the top two expressed sites of the genes.

expressed (supported by ≥10 PASS reads), with an average of 5.0 expressed polyA sites per gene.

We further grouped these putative sites based on their genomic locations (Fig. 4b). A total of 89.1% of the sites are located in introns, while 3.2% are located in terminal exons, and 6.8% are located in downstream extended regions (Fig. 4c). Only 1.5% of intronic sites are expressed (Fig. 4d). By comparison, 39% of terminal exonic sites are expressed, and 6.0% of sites located in the downstream extended regions are expressed (Fig. 4d). Even when expressed, intronic sites show significantly lower usage levels than terminal exonic sites (Fig. 4e and Supplementary Fig. 7d). These data indicated that intronic polyadenylation is generally repressed due to competition with splicing.

### Quantitative modeling of genomic features determining intronic polyA site expression

We next examined genomic features distinguishing used vs. unused intronic sites. Based on the local splicing structures, we classified the putative intronic sites from the PolyaID model into composite and skipped types (Fig. 5a, g). The skipped intronic sites showed splicing between the upstream 5' splice site and the polyA site, while the composite sites did not. For both types, used intronic polyA sites are significantly stronger than unused sites (Fig. 5b, h).

Furthermore, we controlled for strength and identified gene structure features separating used vs. unused polyA sites (Fig. 5c,i; see "Methods" for details). First, the introns containing used composite sites have weaker 5'SS and 3'SS, and larger upstream and downstream exons (Figs. 5c and S8a), indicating that intronic polyadenylation tends to happen when the surrounding splicing efficiency is low. Second, the used sites tend to be located close to the 5'SS (Fig. 5c), presumably because the short distance allows more time for intronic polyadenylation to happen before splicing. Finally, used sites are enriched in the last introns of genes (Fig. 5c). Based on the exon definition model of splicing in mammals[30], the splicing activity of the last intron should be weaker compared to the upstream introns.

For the skipped intronic sites, the introns containing used sites had stronger upstream splicing activity including smaller upstream

exon sizes and stronger upstream 3'SS, and showed weaker downstream splicing strength including larger downstream exon and weaker downstream 3'SS (Figs. 5i and S8b). These data indicated that the competition between two alternative 3' splice sites regulates the intronic site usage. Similar to the composite type, expressed skipped intronic sites were also more likely to be located close to the upstream splice sites and in the last introns (Fig. 5i).

Then we built logistic regression models to classify the used vs. unused intronic sites. The genomic features described above showed significant contributions to the models and adding additional features did not increase the classification accuracy (Fig. 5d, j). The classifiers achieved an AUROC = 0.916 for composite intronic sites, and an AUROC = 0.911 for skipped sites (Fig. 5e, k). We further confirmed the accuracy of our models by analyzing published RNA sequencing (RNA-seq) data, and sites with higher usage probability showed a larger decrease in read density after the polyA site (Fig. 5f, l). Taken together, we showed that the genomic context of intronic polyA sites plays a driving role in determining their expression. We developed quantitative logistic regression models to reveal key genomic regulatory features and to calculate the usage probability of PolyaID-identified putative sites (Fig. 5m, n).

### Examining genomic features regulating terminal exonic polyA site usage

As cleavage and polyadenylation occur co-transcriptionally with RNA polymerase II (Pol II) elongation, the order and spacing between tandem polyA sites in terminal exons is expected to regulate their usage. Here, we aimed to quantify the genomic parameters regulating terminal exonic site usage (Fig. 6a). Using 3'READS, we calculated the relative usage levels of proximal and distal polyA site pairs. As expected, the high usage sites showed stronger PolyaStrength scores (Fig. 6b–d).

Next, we selected proximal polyA sites with comparable absolute and relative strength to the paired distal site and examined other genomic features distinguishing the highly vs. lowly used ones (Fig. 6e). First, when the distances between the two sites were longer,

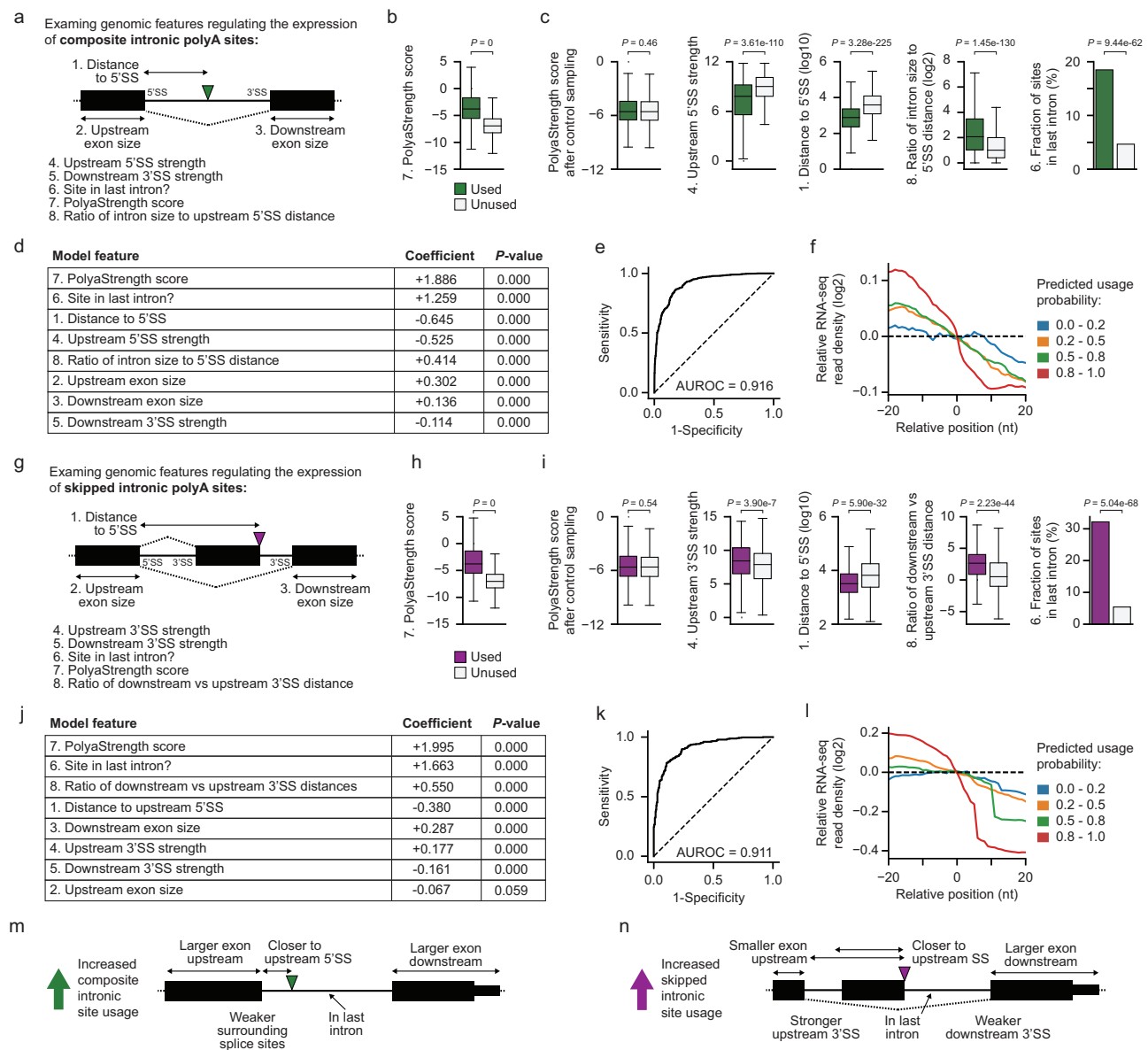

**Fig. 5 | Examining genomic features determining the usage of intronic polyA sites. a** Diagram showing the genomic features of composite intronic polyA sites. **b** Comparison of the PolyaStrength scores for used (≥10 reads, N = 7720) vs. unused (0 reads, N = 7720) composite intronic sites. We sampled an equal number of unused sites for the analyses. The boxes are bounded by the 25 and 75 percentiles and the center represents the median. The whiskers extend from each edge of the box to indicate the 1.5× interquartile range (IQR). Outliers beyond the whiskers were excluded. The two-sided Wilcoxon rank sum test P-value is shown. **c** We controlled for the polyA site strength and compared selected genomic features between used (green) and unused (gray) composite intronic sites (N = 2812 sites per group). See (**b**) for boxplot definition. The two-sided Wilcoxon rank sum test P-values are shown. **d** The modeling features, coefficients, and two-sided t-test P-values from the logistic regression model classifying used vs. unused composite intronic sites. **e** ROC curve showing the performance of our composite intronic logistic regression model. **f** RNA-seq read density around the putative composite

intronic sites, grouped by predicted probabilities. **g** Diagram showing the genomic features of skipped intronic sites. **h** Comparison of the PolyaStrength scores for used (≥10 reads, N = 3284) vs. unused (0 reads, N = 3284) skipped intronic sites. See (**b**) for boxplot definition. The two-sided Wilcoxon rank sum test P-value is shown. **i** We controlled for the polyA site strength and compared selected genomic features between used (green) and unused (gray) skipped intronic sites. (N = 1146 sites per group) See (**b**) for boxplot definition. The two-sided Wilcoxon rank sum test P-values are shown. **j** The modeling features, coefficients, and two-sided t-test P-values from the logistic regression model to classify used vs. unused skipped intronic sites. **k** ROC curve showing the performance of our skipped intronic logistic regression model. **l** RNA-seq read density around the putative skipped intronic sites, grouped by predicted probabilities. **m** Genomic features promoting composite intronic polyA sites usage. **n** Genomic features promoting skipped intronic polyA sites usage.

the proximal sites were more often used (Fig. 6f). This is presumably because cleavage and polyadenylation have more time to happen at the proximal site when the distance between two sites is large. Second, the size of the last intron also affected polyA site choice, and a smaller last intron size promoted the usage of the proximal polyA site (Fig. 6g). These data suggest that efficient splicing of the last intron promotes terminal exonic polyadenylation in proximity. Finally, when the

distance to the downstream antisense gene was short, the proximal polyA site was more often used (Fig. 6h). A possible explanation for this is that if two adjacent genes are located on opposite strands, the Pol II from each strand can collide with each other in the 3′UTRs or extended regions before termination, which results in the decreased elongation rate of Pol II. In this case, the proximal polyA site is more likely to be used[31].

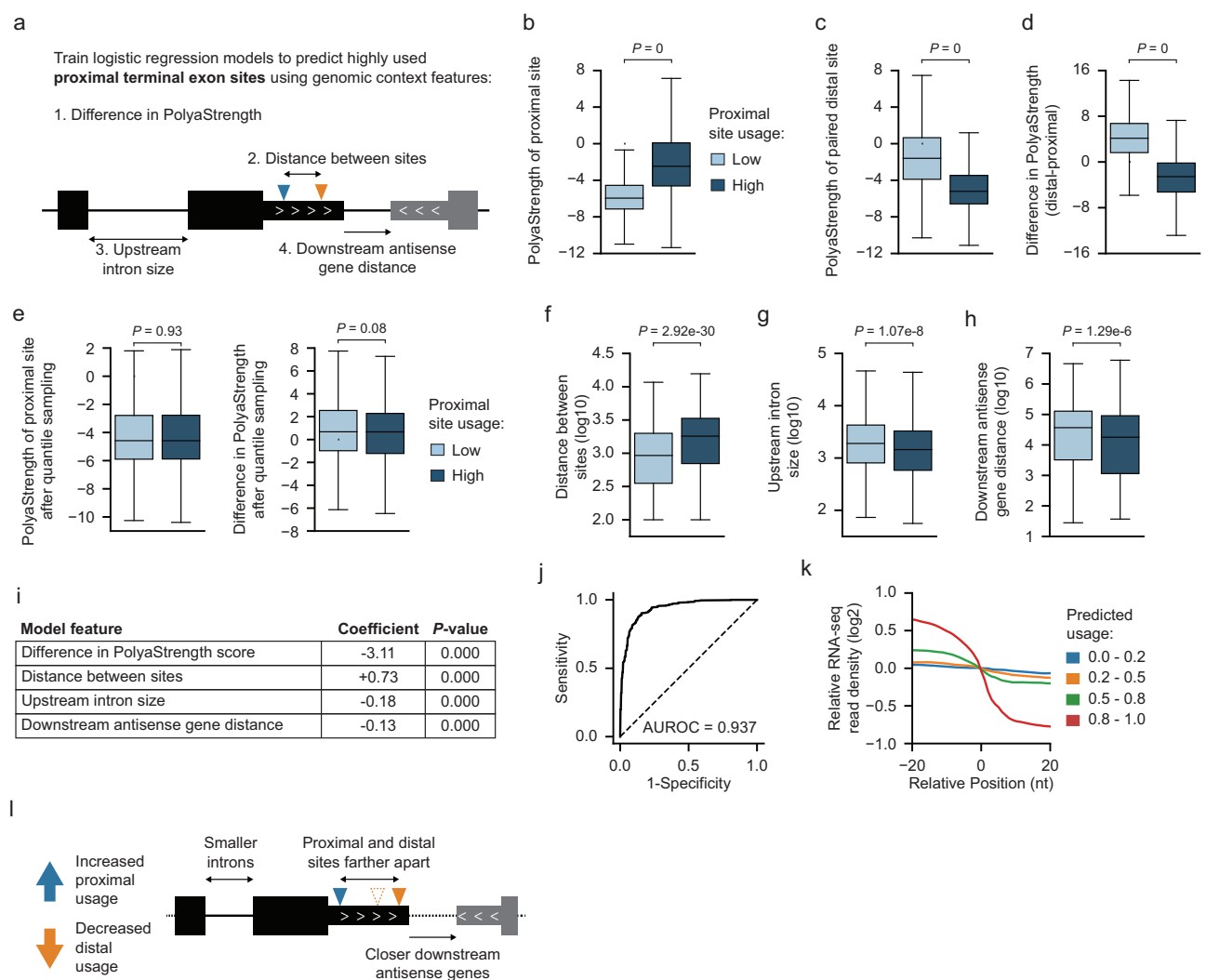

**Fig. 6 | Examining genomic features regulating terminal exonic site usage.**
**a** Diagram showing the genomic features of APA sites in terminal exons. **b**–**d** For the paired terminal exonic sites, we calculated the usage level of a proximal site as the ratio between the read number supporting the proximal site and the read number supporting both proximal and distal sites. We grouped the proximal sites based on their usage levels: high (top 50%, $N = 4721$), and low (bottom 50%, $N = 4722$). Then we compared the PolyaStrength scores of highly and lowly used proximal sites. The boxes are bounded by the 25 and 75 percentiles and the center represents the median. The whiskers extend from each edge of the box to indicate the 1.5× interquartile range (IQR). Outliers beyond the whiskers were excluded. The two-sided Wilcoxon rank sum test *P*-values are shown. **e**–**h** We sampled the proximal polyA sites with comparable strengths as well as the relative strengths vs. distal sites ($N = 1186$ for each group) (**e**). Then we examined indicated genomic features distinguishing highly vs. lowly used proximal sites (**f**–**h**). See (**b**) for boxplot definition. The two-sided Wilcoxon rank sum test *P*-values are shown. **i** We built a logistic regression model to classify highly vs. lowly used proximal polyA sites. The table shows the modeling features, their coefficients and *P*-values from two-sided *t*-tests. **j** The ROC curve showing the performance of our logistic regression model classifying highly vs. lowly used proximal polyA sites. **k** The RNA-seq read density around the proximal polyA sites grouped based on predicted probabilities in (**j**). **l** The schema showing genomic features promoting the usage of proximal polyA sites in terminal exons.

We built a logistic regression model to classify the highly vs. lowly used proximal polyA sites. The features described above showed significant contributions to the classification ($P = 0$; Fig. 6i). The model had an AUROC = 0.937 (Fig. 6j). We also validated the predicted usage using RNA-seq, and higher usage sites showed larger decreases of reads after the polyA sites (Fig. 6k). Taken together, we showed that APA in terminal exons is regulated by local genomic parameters in addition to the site strengths (Fig. 6l).

**Our models revealed disease/trait-associated genetic variants regulating polyadenylation activity**

We next assessed the ability of our PolyaID and PolyaStrength models to predict the effect of genetic variants altering polyadenylation signals. A previous study analyzed GTEx genetics and RNA-seq data and revealed 330 functional PAS SNPs (called PAS pA-QTLs) associated with significant differential usage of the sites across human tissues[32]. We performed PolyaID and PolyaStrength predictions using the reference and alternative allele sequences (Supplementary Data 5). The predicted score changes correctly captured polyA site strengthening when creating a PAS (AAUAAA or AUUAAA) as well as the site weakening upon the PAS loss (Supplementary Fig. 9a, b). We also used this set of PAS pA-QTLs to establish a procedure for identifying variants significantly impacting polyadenylation activity. We required that at least one allele have a PolyaID classification probability ≥ 0.9 and PolyaStrength score ≥ −9 and that the variant leads to a |ΔPolyaStrength| ≥ 1 (Supplementary Fig. 9c–e). This conservative threshold captured nearly 70% of PAS pA-QTLs as significant.

Using our deep learning models, we characterized the impact of disease/trait-associated variants annotated by ClinVar[33], the UK BioBank[34], and GWAS Catalog[35] on polyadenylation activity. If a known disease/trait-associated variant impacted any of the *cis*-regulatory

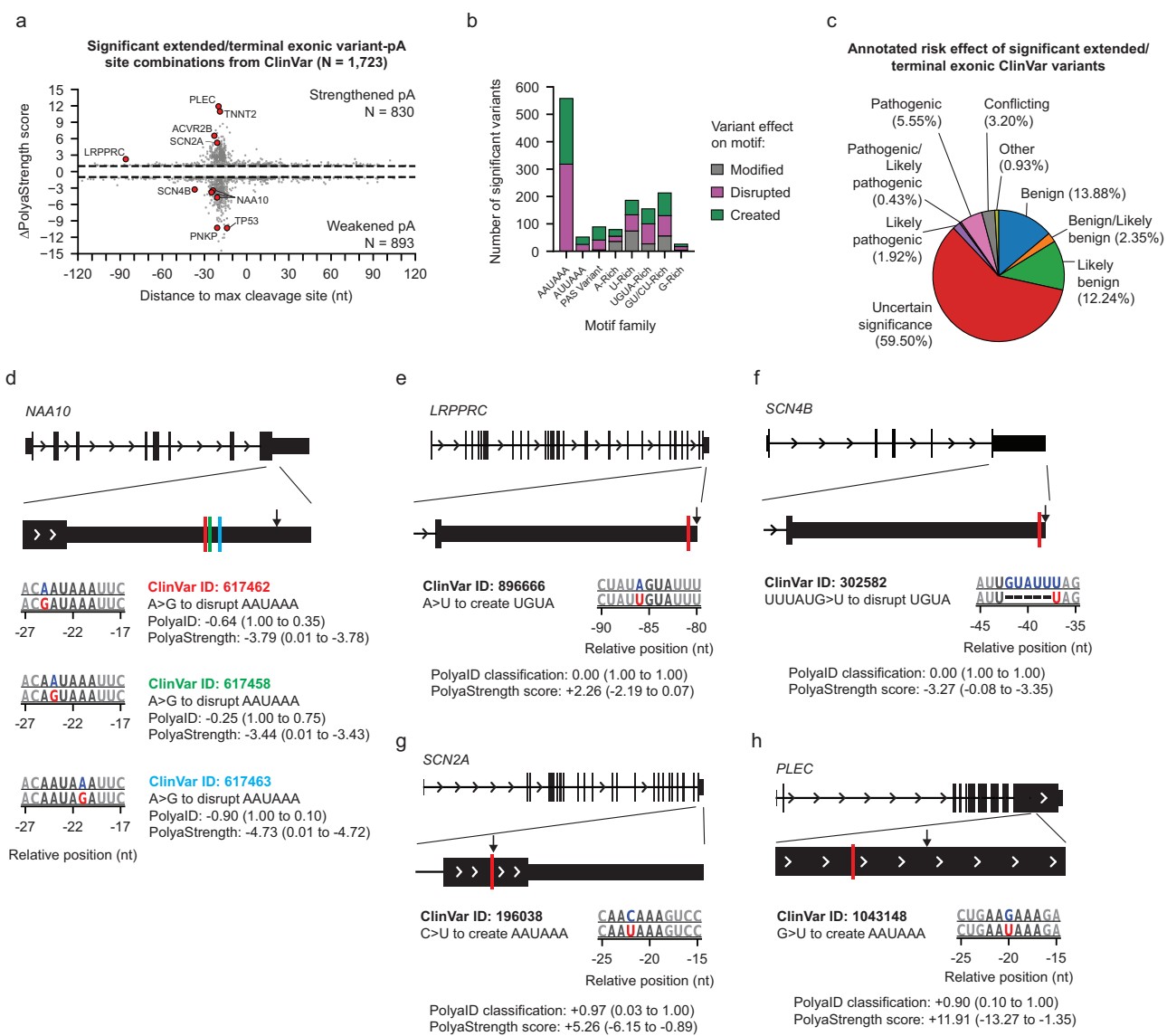

**Fig. 7 | Identifying disease/trait-associated genetic variants regulating polyadenylation activity in 3′ terminal exons. a** Effects of ClinVar variant on PolyaStrength score versus the distance to the maximum cleavage site. Points represent each variant test location relative to the polyA site. **b** Frequency of motif families affected by terminal exonic ClinVar variants. Bars colored by whether the motif of interest was disrupted, modified, or created as a result of the mutation. **c** The annotated risks of disease-associated ClinVar variants altering polyadenylation activity. **d–g** Example variants altering polyadenylation activity. For each variant, we present its ID, affected motif type, the PolyaID classification score changes, and the PolyaStrength score changes.

elements that we identified (Fig. 2), we used the PolyaID model to scan the 240 nt region surrounding the variant and calculated the probability of the reference and alternative allele sequences being a polyA site. If the variant induced changes in PolyaID classification probability and PolyaStrength scores meeting the three criteria defined above, we used the PolyaID cleavage profile to obtain the representative polyA site location and calculated the PolyaStrength scores to measure the change in strength caused by the mutation.

We first analyzed the genetic variants located in terminal exons. Using the criteria we established above analyzing PAS pA-QTLs, we found 1723 ClinVar, 1045 UK BioBank, 87 GWAS variants with altered polyadenylation activity (Fig. 7a, b, Supplementary Fig. 11a–f, and Supplementary Data 6). Half (48%) of the variants increased the site strength, while the remaining half weakened the sites. 60% of ClinVar variants were annotated as "uncertain significance" and their regulatory impacts on polyadenylation activity shown here could be the molecular mechanism mediating their association with diseases/traits (Fig. 7c).

Our models identified variants disrupting the AAUAAA signal sequence and drastically weakening the polyadenylation activity in genes such as these in *TP53*, *BRCA1*, *NAA10*, *PNKP*, *HBA2*, and *INS* (Fig. 7d, Supplementary Fig. 10b–c and Supplementary Data 6). Some of these variants were previously experimentally characterized to be disease-causing[12–14]. We also revealed the genes showing PolyaStrength changes by other regulatory elements. For example, the genes *LRPPRC* and *SCN4B* contain variants affecting the UGUA motif which were predicted to impact polyA site strength (Fig. 7e, f).

While previous genomic analyses of polyA site variants were limited to the expressed sites annotated by databases[19,24,36,37], our model can reveal de novo polyA sites caused by genetic mutations. For example, the loss-of-function of the sodium voltage-gated channel alpha subunit 2 (*SCN2A*) is a driver of neuronal disorders such as epilepsy[38], and the gene contained a synonymous variant (ClinVar ID: 196038, AAC > AAU) in the coding region of the terminal exon (Fig. 7g). This variant created an AAUAAA signal resulting in the generation of a de novo downstream polyA site. The expression of this polyA site

would generate RNA isoforms without producing full-length protein. This could be a new mechanism causing the downregulation of SCN2A protein in epilepsy patients.

Similarly, the gene *PLEC* (encoding the protein plectin) contains a missense variant in the coding region (ClinVar ID: 1043148, AAG > AAU) of its terminal exon and creates the PAS AAUAAA. This variant is associated with multiple diseases, such as muscular dystrophy, nail dystrophy, and skin disorders[39–41]. The expression of the de novo polyA site would generate the non-stop codon RNA isoform, which cannot produce the full-length protein (Fig. 7h). For the gene *TNNT2*, a synonymous variant was located at the stop codon (ClinVar ID: 1106650, UAG > UAA) and created an AAUAAA PAS, which is predicted to form a de novo polyA site downstream (Supplementary Fig. 10d). Expression of this site would generate an mRNA isoform with a shorter 3′UTR compared to the wild type. Finally, a variant (ClinVar ID: 344948, UAGAGAA > U) in the gene *ACVR2B* created an AAUAAA sequence and a downstream polyA site (Supplementary Fig. 10e).

We also performed predictions for the genetic variants located in introns. As the intronic site expression is tightly controlled by the splicing environment, we further required that the sites showed a predicted expression probability ≥0.9 using the logistic regression models we developed in Fig. 5. In total, we identified 700 ClinVar, 1098 UK BioBank, 113 GWAS variants predicted to alter polyadenylation activity and be expressed (Supplementary Fig. 12 and Supplementary Data 6). Taken together, these results showed that our models can be used to reveal genetic variants regulating polyadenylation activity, which may contribute to human disease progression or trait formation.

## Discussion

In this study, we developed a series of deep/machine learning models that can be applied in concert to study polyA site definition, strength, cleavage, and usage in the genomic context. We first developed the PolyaID model to classify the polyA site sequences vs. random nucleotides. Due to our unique model design, PolyaID is not restricted to certain PAS types and can scan the genome sequence to unbiasedly identify putative polyA sites at nucleotide-level resolution. This cannot be achieved by previously published methods. Next, we developed the PolyaStrength model to calculate the sequence strength of polyA sites. Then we used logistic regression to reveal the local gene structure parameters that determine site expression in introns and terminal exons. The models provide a unique computational toolkit to characterize genome-wide polyadenylation regulation.

Using the PolyaID and PolyaStrength models, we quantitatively measured the positional interactions among *cis*-regulatory motifs. Our results revealed the optimal tandem motif positions for AAUAAA and other signal variants recognized by the CPSF complex (−21 nt), UGUA recognized by CFI (−37 nt), U-rich elements bound by FIP1 (−15 nt), GU/CU-rich (+10 nt) and U-Rich motifs (+18 nt) bound by CstF, as well as G-rich elements bound by hnRNPs (+37 nt). Small distance changes between the motif pairs (<10 nt) can result in differential importance scores, indicating that subtle variance in subcomplex configuration regulates polyadenylation activity. A published biochemical study focused on positional crosstalk between UGUA and AAUAAA elements[27] and showed that UGUA showed the strongest activity promoting polyadenylation when it is located a short distance (20 nt) upstream of AAUAAA. This finding confirms the accuracy of our analysis. Our modeling is not restricted by designed reporters but quantifies all positional possibilities among the motifs without labor-intensive biochemical assays. The characterization of positional effects among motif pairs may enhance the structural understanding of polyadenylation complex formation.

Studies showed that RNA isoforms generated by heterogeneous cleavage at a polyA site can show quite variable stability levels due to the regulation of RNA secondary structure formation, even when they

only have a few nucleotide differences at the 3′-end[42,43]. Here we showed that the cleavage heterogeneity of the polyA sites is regulated by two different mechanisms: (1) the number of PAS AAUAAA; and (2) the relative distance between the PAS AAUAAA and downstream GU/CU- or U-rich elements. A shorter distance between CPSF and CstF binding can lead to a more uniform/stable complex configuration and results in more consistent cleavage by the endonuclease CPSF73[44]. Our PolyaID model captured this regulatory mechanism and can be used to predict and control the cleavage events when engineering polyA sites.

Importantly, our PolyaID model achieved unbiased identification of genome-wide putative polyA sites at nucleotide resolution for the first time. More than three million putative sites were identified across gene regions based on the nascent RNA sequences with an average of 2 sites per kb. As introns are 30-fold longer than exons, 89.1% of putative polyA sites are located in introns. 98.5% of these intronic sites were not expressed, as aberrant expression of these sites could generate truncated proteins. The optimization of splicing signals is crucial for suppressing intronic polyadenylation[15]. Previous studies showed that disrupting the splice sites or using anti-sense oligonucleotides to block U1 snRNP recruitment can increase intronic polyadenylation[16,45,46]. These results are in line with our findings of the competition between splicing and intronic polyadenylation. Here, we developed quantitative models to capture the regulatory parameters and calculate the probability of a site being used.

Although intronic polyadenylation is generally unfavored, some genes express functional RNA isoforms using this mechanism[47]. For example, the *CSTF3* gene contains an intronic polyA site associated with a weak 5′SS conserved across vertebrates, which plays fine-tuning roles in full-length protein expression[16]. The immunoglobulin M gene uses intronic polyadenylation to generate two RNA isoforms encoding membrane-bound and secreted proteins with different C-terminal domains[48]. It will be interesting to apply our models to examine the coevolution of intronic polyadenylation and surrounding gene structures across species, which may distinguish the functionally conserved vs. randomly occurring sites.

APA in 3′-terminal exons regulates the occurrence of *cis*-regulatory elements in 3′UTRs, such as binding sites for microRNAs and RNA binding proteins. This regulation fine-tunes RNA stability and translation efficiency across biological conditions. Besides polyA site strength, our quantitative modeling showed that site usage is correlated with the tandem site distance, the size of the last intron, and the distance to the next downstream antisense gene. Our model achieved the classification of highly vs. lowly used terminal exonic sites with an AUROC = 0.937. The small fraction of misclassification could be attributed to the regulation of RNA stability or cell type-specific regulation. Our models were trained mostly based on mature RNA expression, although a few 4sU-labeled nascent RNA libraries were included. The principles we learned from this study represent general polyadenylation regulation.

Future work integrating (nascent) RNA-seq data with our PolyaID/PolyaStrength models could comprehensively identify APA events and reveal molecular mechanisms underlying biological process- or cell type-specific regulation. Using the 3′READS data, we examined APA regulation across five different cell types, including cerebellum tissue, neuronal stem cells (NSCs), embryonic stem cells (ESCs), as well as two placental cell lines HTR8 and JEG3. Using principal component analysis of sites showing significant APA regulation across the cells/tissues (see "Methods" for details), we observed that cerebellum and NSCs (named as neuronal-like cells) show similar APA profiles, while ESCs, HTR8, and JEG3 cells (named as embryonic-like cells) were clustered together (Supplementary Fig. 13a). Consistent with previous reports[5,49], more genes showed higher usage of proximal polyA sites in embryonic-like cells than neuronal-like cells (1409 vs. 226 genes) (Supplementary Fig. 13b). Proximal polyA sites showing higher usage in embryonic-like cells tend to have lower PolyaStrength scores and be located far away

from the distal sites, compared to those with higher usage in neuronal-like cells (Supplementary Fig. 13c–e). These results showed that our deep learning models can be useful for the future investigation of cell-type specific APA regulation.

We applied our models to examine the impact of disease/trait-associated genetic variants affecting polyadenylation activity. Most published methods studying APA genetics were limited to annotated polyA sites from curated databases. Our models are not restricted by existing site annotation, PAS types, or repetitive genomic regions. They can identify de novo polyA sites caused by the mutations and expand our ability to tackle human genetics controlling polyA site usage. Our analyses effectively identified known mutations driving disease progression, and also revealed variants causing de novo polyA site formation. Our catalog of polyadenylation-altering variants provides a useful resource for future experimental characterization of polyadenylation variants mediating human disease progression or trait formation.

## Methods

### Genome-wide polyA site identification and annotation using 3'READS data

To quantify genome-wide polyA site expression, we analyzed the 3'READS data from 103 human samples (Supplementary Data 1), including 19 primary tissues and cell line types. We first trimmed the leading Ts from the sequencing reads and recorded the number of Ts. Trimmed reads were then aligned to hg38 using TopHat v2.1.0[50] with the gene structure annotation from the GENCODE database[51]. The uniquely mapped reads with ≥2 non-genomic As were considered PASS reads. The PASS reads revealed cleavage activity at the nucleotide resolution.

To account for the imprecision inherent in cleavage and polyadenylation, we clustered individual sites within 24 nt and selected the position with maximum PASS reads as the representative site. Terminal exonic polyA sites can be located downstream of annotated 3' ends. We extended the 3' end by a maximum of 5 kb, but not overlapping with the downstream gene transcription start site on the same strand. Because we merged many samples for these analyses, to minimize false positives, we required that retained polyA sites should be supported by ≥10 PASS reads and that their expression relative to the total PASS reads in the same gene should be >2% in at least one cell/tissue lineage. In total, we analyzed 1.5 billion sequencing reads and identified 109,814 polyA sites (Supplementary Data 2).

### Developing the PolyaID model

**Input datasets.** To generate positive polyA site sequences for the model training, we randomly sampled 20,000 sites weighted by the number of supporting reads across the genome. For each site, we included the 240 nt sequences centered at each position between −25 and +25 nt surrounding the representative cleavage site as positive examples. For negative control sequences (240 nt), we used random intergenic sequences, shuffled transcript sequences, and neighboring sequences 50–100 nt distant from a polyA site that did not contain any PASS reads in the middle 50 nt regions. With these site-neighboring sequences in the negative examples, our algorithm only predicts positive polyA sequences showing cleavage in the middle region.

Our PolyaID model contains two output branches: the classification probability and the cleavage probability profile. For model training, the classification score for positive sequences was 1, while negative sequences used the score 0. We used 3'READS within +/−25 nt of the positive site to represent the cleavage vector (length = 50). This vector sum is normalized to 1 to show the cleavage probability at each position. Negative controls were assigned a constant cleavage probability vector at a background value of 0.02.

**Model architecture and training.** For the input to the PolyaID model, we converted the 240 nt sequences (positive and negative examples)

into a (4 × 240) matrix using one-hot encoding. Each column of the matrix represented a position in the sequence, and each row corresponded to the nucleotide A, C, G or U. A 1 was marked if the nucleotide was present in the sequence position, and all others were marked with 0. This creates a numerical representation of the sequence fed to the model. Our sequence-modeling unit was composed of a convolutional layer and a bidirectional LSTM layer. The window size for learning the sequence composition was set to 8. The output layer contains two units: the classification probability calculating the likelihood a sequence is a polyA site, and the cleavage probability vector (length = 50). Details of the layers used in this model are described in Supplementary Data 3.

We use the binary cross-entropy loss function for the classification branch (Eq. 1):

$$Loss_{Classification} = -\frac{1}{N}\sum_{i=1}^{N}\left(y_i{}^*\log(p(y_i)) + (1-y_i){}^*\log(1-p(y_i))\right)$$

where $y_i$ is the label for the sequence $i$ (1 for positive site and 0 for negative sequence) and $p(y_i)$ is the predicted probability of a sequence $i$ being a polyA site.

We used the Kullback-Leibler divergence loss function for the cleavage branch (Eq. 2):

$$Loss_{Cleavage} = \sum_{j=1}^{50}\left(c_j{}^*\log\left(\frac{c_j}{o_j}\right)\right)$$

where $c_j$ and $o_j$ are the predicted and observed cleavage probabilities, respectively, for each position $j$ across the 50 nt window vector.

The loss functions were implemented out-of-box in TensorFlow2. Data were fed into the model in batch sizes of 100 with a learning rate of 0.001 and optimized using the Adam optimizer with Nesterov momentum.

The dataset was randomly partitioned into 80% training, 10% validation, and 10% holdout test splits, with positive and negative controls balanced. The loss functions for each of the output branches were equally weighted during training. Training was monitored using the loss function values and the accuracy of predictions. Several models were trained in replicate with different random initializations, and the most representative model was chosen. The final model was trained for four epochs, after which it attained the minimum loss on the validation set.

**Model evaluation.** We assessed the quality of our trained model using the performance on a holdout test set. Based on PolyaID prediction, we calculated the classification probability as well as the cleavage probability for each site. Classification performance was shown using the AUROC and AUPRC from the Python library scikit-learn.

The PASS reads from 3'READS data revealed near single-nucleotide resolution of cleavage sites with one limitation. We cannot distinguish the last nucleotide As from genome sequences vs. those generated by polyadenylation. Actually, many cleavage events tend to happen within A-rich regions or after a CA dinucleotide[26]. If there are multiple genomic As at the end of mapped reads, here we assigned the reads to the first A. Our approach is different from the APARENT model where they assigned the reads to the last non-A site. As a result, the cleavage site predicted by the APARENT model does not happen at the As, while our PolyaID model predicts the cleavage probability among the As.

For the cleavage probability vector, we subtracted the background probability of 0.02 from the predicted vector and then renormalized the distribution to the sum of 1. The performance of cleavage probability vector prediction was assessed by comparing the observed and predicted mean cleavage positions (MCP), which were calculated from the following dot product.

For position $i$ from a vector (length = 50), suppose the cleavage probability is $p_i$. The MCP $m$ was calculated as follows (Eq. 3).

$$m = \sum_{i=1}^{50} (p_i \times i)$$

The MCP metric determines the most likely cleavage position weighted by the cleavage probabilities in that region and quantifies the agreement between the shape of observed probabilities from the 3'READS data and predicted probabilities from our model.

### Predicting genome-wide putative polyA sites using PolyaID at single-nucleotide resolution

Using a scanning window across the genome with one nucleotide per step, we performed a PolyaID prediction using the 240 nt nascent RNA sequence. As shown in Fig. 1f and Supplementary Fig. 2g, around a polyA site, we generally obtained a broad region (~60 nt) considering high classification probability scores alone. To achieve the nucleotide-resolution prediction, we consolidated the PolyaID-predicted classification probability and the cleavage probability vector to infer the locations of putative polyA sites. We required classification probability >0.5 and the center of the cleavage probability vector >0.05 to create a subset of positive sequences showing cleavage in the middle. We overlayed the cleavage probability vectors from these positive predictions and took the position-wise mean. The location with maximum cleavage probability within a positive cluster was used to represent the putative cleavage and polyadenylation site.

We further grouped the putative polyA site based on their genomic locations: intronic, upstream exonic, terminal exonic, and extended terminal exonic sites. We examined whether these polyA sites are expressed (supported by ≥10 PASS reads). To compare the relative usage levels of the site types, we selected genes showing ≥2 expressed sites in terminal exons or extended regions and calculated the usage level of a site as the ratio between the number of PASS reads supporting the site vs. the sum read counts of the top two expressed sites in the (extended) terminal exons.

### Developing the PolyaStrength model

**Input dataset.** We used the relative expression of APA sites located in terminal exons to represent their usage levels. The relative usage of a polyA site is calculated as the ratio (log-odds transformed) between the number of reads supporting the site and the summed reads from the highest and second-highest expressed sites in the same gene. We did not use the total read number from a gene as the denominator because there are different numbers of expressed polyA sites across genes.

Supposing the supporting read number for site $i$ is $n_i$, its relative usage $u_i$ and the log-odds transformed value $o_i$ were calculated as (Eq. 4):

$$u_i = \left( \frac{n_i}{n_{max\,site} + n_{second\,max\,site}} \right)$$
$$o_i = \log 2 \left( \frac{u_i}{1 - u_i} \right)$$

**Model architecture and training.** The polyA sequences (240 nt) were converted into a (4 × 240) matrix using one-hot encoding, the same as for the PolyaID model. For the Sequence-Modeling unit, we also used a convolutional layer followed with a bidirectional LSTM layer, but their parameters are different from those of PolyaID and are described in (Supplementary Data 3). The output layer predicts the log-odds transformed usage levels. We used the mean squared error

implemented in TensorFlow2 as the loss function for model training (Eq. 5):

$$Loss_{usage} = \frac{1}{n} \sum_{i=1}^{n} (u_i - v_i)^2$$

where $u_i$ and $v_i$ are predicted and observed usage level for the polyA site $i$. Data was fed into the model during training in a batch size of 100, with a learning rate of 0.001, and optimized using the Adam optimizer with Nesterov momentum.

53,105 terminal exonic polyA sites were used to build the model. And we required that the selected genes should contain ≥2 polyA sites in terminal exons. The sites were randomly separated into 80% training, 10% validation, and 10% holdout test splits. Training was monitored using the loss function and the accuracy of predictions. The model was trained for 9 epochs, after which it attained the minimum loss on the validation set. Several models were trained in replicate, and a representative model was chosen.

**Model evaluation.** We grouped polyA sites in the testing set based on their relative usage levels and examined whether they showed differential strengths. As one gene can express multiple stronger/weaker sites, we did not expect their usage levels to be perfectly correlated with the predicted strengths. To further examine the accuracy of our model, we selected 8,167 pairs of APA sites in terminal exons showing >8-fold usage level difference. We calculated the AUROC using the PolyaStrength score to distinguish highly vs. lowly used sites, as shown in Fig. 1i.

### Comparison with published models

We compared the performance of PolyaID vs. previously published models (i.e. DeepPASTA[18], APARENT[19], APARENT2[20], and PolyApredictors[21]) in classifying polyA sequences vs. random nucleotides. We split the sites from the PolyaID holdout test dataset based on their PAS types: AAUAAA, AUUAAA, or other variants, respectively. And we randomly sampled an equal number of negative controls to balance. The same positive and negative datasets were used to compare the performance among the models. We calculated the AUROC and AUPRC values using the predicted PolyaID classification probability, DeepPASTA classification probability, APARENT/APARENT2 isoform abundance, and PolyApredictors expression to compare classification performance. In Supplementary Fig. 2b–c, the polyA sequences were centered around the highest probability cleavage site to make predictions. In Supplementary Fig. 2d–e, we randomly selected 10,000 polyA sites with each PAS, shifted these sites from −25 to +25 nt around the maximum cleavage sites and used the shifted sequences to make the predictions. To assess the model predictions of cleavage probabilities, we calculated the Pearson correlation between the observed and predicted mean cleavage position for positive polyA sites from PolyaID, APARENT, APARENT2, and PolyApredictors, shown in Supplementary Fig. 2f. Individual examples were shown in Supplementary Fig. 2k–l.

To compare models to distinguish highly vs. lowly used sites, we made predictions using the sequences surrounding the set of 8167 pairs of APA sites in terminal exons showing >8-fold usage level difference. We then calculated the AUROC and AUPRC using the PolyaStrength score, APARENT and APARENT2 isoform abundance, and PolyApredictors expression predictions, which are shown in Supplementary Fig. 3a–c.

Our above analyses were based on the polyA sites defined in this study. We also evaluated the model performance using polyA sites defined in published databases including PolyA_DB v3[24] and APADB v2[25], as well as the site expression from a massively parallel reporter assay (MPRA)[19]. PolyA_DB sites were also defined by 3'READS. We selected 36,701 sites supported by mean reads per million (RPM) ≥3 as

positive examples and the same number of random nucleotide sequences as negative examples to evaluate the performance of site classification using PolyaID, DeepPASTA, APARENT, APARENT2, and PolyApredictors. We picked 6,871 terminal exonic polyA site pairs with baseline expression RPM ≥1.5, with ≥8-fold expression difference, and ≥100 nt apart to evaluate the performance of PolyaID, APARENT, APARENT2, and PolyApredictors in classifying the strong vs weak polyA sites. For the APADB model, the polyA sites were defined by massive analysis of cDNA ends sequencing (MACE reads). We selected 48,860 sites supported by ≥10 MACE reads as the positive examples for evaluate the site classification, and used 8,000 paired terminal exonic polyA sites with baseline expression ≥5 MACE reads, with ≥8-fold expression difference, and ≥100 nt apart to evaluate the performance of classifying strong vs. weak sites.

We also analyzed a published MPRA dataset[19] in which they introduced hundred thousands of random nucleotides into the polyA site sequence, and quantify the site expression using deep sequencing. The relative transcript expression levels can be used to measure the relative strength of the polyA sites. We analyzed the "HSPE1" library which was used for their model evaluation in the published paper. To remove the false-positive sites from oligo(dT) internal priming, we only retained sequences with PolyaID-predicted classification probability ≥0.5 for further analyses. We grouped 337,044 sequences into 5 bins based on their supporting read numbers, and evaluated the performance of the algorithms to classify the top highly vs. lowly expressed (group 5 vs. group 1) polyA site sequences.

## Characterizing *cis*-regulatory elements mediating polyA site identification and strength

We employed a motif disruption approach to systematically quantify the importance of a hexamer to the PolyaID and PolyaStrength prediction for a polyA site sequence. For each hexamer occurrence in a polyA site sequence, we replaced it with randomized nucleotides 100 times while keeping the remaining sequences the same. We performed predictions for both the original and modified sequences. Then the effect of motif disruption was quantified using the median change in the log-odds of the PolyaID-predicted classification probability or PolyaStrength score from the 100 replacements.

To identify motifs driving genome-wide polyA site formation, we performed the above-described analyses for 32,682 well-expressed polyA sites with ≥100 PASS reads and usage levels ≥5%. The usage level of a polyA site was calculated as the ratio of PASS reads supporting the site vs. the total PASS reads assigned to the gene. For each position from −120 nt to +114 nt surrounding polyA sites, we calculated the sum importance score of a hexamer in that position as the sum change in log-odds classification probability or strength. We also calculated per site importance score for each hexamer by dividing the sum score by the frequency.

We next analyzed *cis*-regulatory elements contributing to genome-wide polyA site identification, including AAUAAA and PAS variants[22], UGUA, U-rich, GU/CU-rich, and G-rich motifs. U-rich elements were defined as those with 5 Us in a hexamer, with a U required in the first position. G-rich elements were defined as those with 5 Gs in a hexamer, with a G required in the first position. GU/CU-rich hexamers contained GUGU, UGUG, GUCU, UGUC, CUGU, or UCUG. We reasoned that the vast majority of motifs located >40 nt upstream or downstream of polyA sites are not important for polyA site formation. We used the sum importance scores from these regions as the background values. The 99.99th percentile score from this background distribution was used as the false discovery rate threshold. The hexamers passing this cutoff in any 40 nt window using either PolyaID or PolyaStrength importance were considered important for genome-wide polyA site formation. We merged motifs recognized by components of the polyadenylation machinery into families for further analysis.

We next examined the differential compositions of *cis*-elements around genomic polyA sites. To this end, we focused on analyzing 13,876 well-expressed polyA sites (supported by ≥100 reads and usage level ≥5%) with an AAUAAA 10-30 nt upstream and either a U-rich or GU/CU-rich motif within 30 nt downstream of the max cleavage site. For these sites, we then determined if they contained any optimally positioned motifs: UGUA bound by CFI 0-40 nt upstream of AAUAAA, U-rich motifs bound by FIP1 between the cleavage site and AAUAAA, G-rich elements recognized by hnRNP F/H within 50 nt downstream of the CstF binding site, or a second CstF motif. We analyzed the frequency of motif combinations, their PolyaStrength scores, the corresponding gene expression calculated by summed 3'READS of the genes, and their site type (i.e. first, middle, last site of APA genes, or single site of a gene). We performed gene ontology analyses using the genes containing sites with ≥5 optimally positioned motifs using the DAVID program[52].

## Characterizing the polyA site cleavage heterogeneity

We used the entropy value to quantify the 3'READS-observed and PolyaID-predicted cleavage heterogeneity. For the polyA sites identified by 3'READS, we selected highly expressed sites supported by ≥100 reads in order to reliably quantify their cleavage distribution. We obtained the 3'READS distribution from the −25 to +24 nt surrounding the representative cleavage site. Supposing the total read number from the polyA region is $n$ and the read count in position $i$ is $m_i$, we calculated the fraction of reads $p_i = m_i/n$. For the PolyaID prediction, we used the predicted cleavage probability $p_i$ for position $i$. We then calculated the entropy value $E$ using the cleavage vector distribution as follows (Eq. 6):

$$E = -\sum_{i=1}^{50} p_i \cdot \log(p_i)$$

The entropy values were used to separate sites with very clear cleavage sites (more "spike"-like, low entropy) and those with heterogeneous cleavage (more nonzero probability sites, high entropy).

We next examined the regulatory motifs driving the differences between high vs. low entropy sites. The entropy values calculated using 3'READS data and PolyaID-predicted cleavage were well correlated. We selected sites that were in the top and bottom 20% of entropy values for both the observed and predicted cleavage profile for the downstream analyses. We divided the polyA sequences into four different regions: −120 - −31 nt; −30 - 0 nt; 1 - 30 nt; and 31 - 120 nt. For each region, we compared *cis*-regulatory element occurrence differences between the low vs. high entropy groups using the Chi-squared test. We also examined the motif enrichment by comparing the fraction of sites containing the motif of interest in each region using the two-sided two proportions test in the low vs. high entropy groups.

Our above motif analyses showed that the distance between upstream AAUAAA (CPSF binding) and downstream U-rich or GUCU-rich elements (CstF binding) regulates the cleavage heterogeneity. We examined whether the PolyaID model could capture this regulation. To this end, we moved AAUAAA and U-rich or GU/CU-rich motifs farther apart or closer together by sliding them through the surrounding sequence, which was unchanged, and predicted the change in polyA site classification probability and cleavage profile. From sites containing AAUAAA upstream of the cleavage site and either U-rich or GU/CU-rich motifs downstream of the cleavage site, we selected those sites in the low entropy group where the motifs were close together (20-40 nt) or in the high entropy group where the motifs were farther apart (40-100 nt). The number of polyA sites in the analyses: moving AAUAAA−U-rich farther $N = 676$, moving AAUAAA−U-rich closer $N = 635$, moving AAUAAA−GU/CU-rich farther $N = 949$, and moving AAUAAA−GU/CU-rich closer $N = 658$. The increase of AAUAAA and

U-rich (or GU/CU-rich) element distances resulted in higher entropy values, and the decrease of distances resulted in lower entropy values.

To validate the genomic parameters regulating polyA site cleavage heterogeneity, we analyzed the MPRA data[19]. As the study reported the sequencing reads supporting each cleavage position of a sequence, we used the distribution of these reads to calculate the entropy value to quantify the cleavage heterogeneity based on the method we used for the 3′READS analyses. We required the sites should be supported by ≥10 reads and retained only the sequences with PolyaID classification probability ≥0.5 to remove false positives from the Oligo(dT) internal priming. To examine the effect of multiple PAS regulating cleavage heterogeneity, we compared the entropy values of sequences with 1 AAUAAA vs. 2 located upstream of the maximum cleavage sites. Because there are many more sequences with only 1 AAUAAA vs. 2, we randomly sampled an equal number of sites with 1 AAUAAA to balance the data for comparison. To examine the impact of the distance between PAS and the CstF binding site in regulating cleavage heterogeneity, we analyzed the sites from the "Simple" and "AARS" libraries with only 1 AAUAAA. We selected these two libraries for analyses because they contain randomly located downstream U-rich or GU/CU-rich elements, while the distance between the PAS and CstF was fixed in other libraries. We grouped the sites based on the distance between the AAUAAA PAS and the first downstream U-rich or GU/CU-rich elements for comparisons.

### Developing logistic regression models to calculate the usage probability of intronic polyA sites

Intronic polyadenylation is generally suppressed because of its competition with splicing. We developed a logistic regression model to determine whether an intronic site would be used based on the surrounding genomic context. Based on the gene structure, we classified intronic sites into two categories: composite and skipped types. The skipped intronic sites show splicing between the upstream 5′SS and the polyA site, while the composite sites do not. For each polyA site, we annotated the following features: polyA site strength, distances to nearby splice sites, surrounding 5′SS and 3′SS strengths, sizes of upstream exon, downstream exon and intron, or whether the site is located in the last intron. The detailed calculation is shown in Supplementary Table 1.

To characterize the impact of genomic features other than site strength regulating intronic site expression, we controlled for site strength using quantile sampling. PolyA sites were split into equally sized bins based on their PolyaStrength scores and then the same number of used and unused sites were sampled from each bin. This equalizes the variance in PolyaStrength between used and unused sites, removing the influence of strength on other genomic features we are investigating.

To train the logistic regression model, we used the genome-wide identified intronic polyA sites as positive examples, and an equal number of negative sites with no supporting reads were randomly selected. For composite and skipped intronic sites, we split the data into training (90%) and testing (10%) sets and then fit independent logistic regression models with the noted features. We used the Python package "scikit-learn" to train the model. The models were evaluated using the AUROC for the testing set to assess the accuracy of the predictions. We tried to incorporate other genomic features into the models (e.g., intron order and strength of splice sites located further away), but those features did not improve the prediction. Our final models only included genomic features contributing significantly to the model with P-value < 0.05. We also further validated these results using RNA-seq density of surrounding sites, binned by predicted usage probability, using data from the ENCODE Project[39,40] (https://www.ncbi.nlm.nih.gov/bioproject/PRJNA30709). We showed the representative results using the RNA-seq from A172 cells, and we obtained

the consistent results from other cell types (i.e. K562, A673, and MCF10A).

### Developing a logistic regression model to examine the relative usage of terminal exonic polyA sites

To learn the genomic features determining the relative usage of terminal exonic polyA sites, we focused on genes with multiple polyA sites in the 3′-most exon. We selected the strongest polyA site per gene and randomly picked another site in the same gene showing ≥8-fold lower usage than the strongest one. Two sites in each gene were categorized as proximal (upstream) and distal (downstream) based on their relative position. In total, we selected the 9,443 site pairs. For terminal exonic APA sites, we annotated the following features: polyA site strength, distance between two sites, distance to the downstream gene in the opposite strand, and size of the last intron. The detailed calculation is shown in Supplementary Table 2.

Based on the above-described features, we developed a logistic regression model to predict whether the proximal or distal polyA site would show higher usage. We split the data into training (90%) and testing (10%) sets. These models classified the individual proximal sites in each pair to determine if it is the more highly used one based on the features described above. We use the Python package "scikit-learn" to train the model. We assessed the performance of these models by plotting the AUROC for the testing set. We only included significant features (P < 0.05) in our final models. We also validated our usage predictions by comparing the RNA-seq read density of surrounding the sites, grouped by the predicted usage probability, using data from the ENCODE Project (https://www.ncbi.nlm.nih.gov/bioproject/PRJNA30709). We showed the representative results using the A172 RNA-seq data and obtained consistent results from other cell types (i.e. K562, A673, and MCF10A).

### Analyses of PAS pA-QTL variants

To evaluate the performance of our models in identifying functional genetic variants, we performed the PolyaID and PolyaStrength predictions of a set of 330 PAS pA-QTLs[32]. These variants were shown to lead to APA regulation by analyzing GTEx RNA-seq data across human tissues. For each site, we obtained its surrounding 240 nt sequences based on the annotation and anchored the PAS motif at −20 nt. We made predictions for both the reference and alternate allele sequences. We grouped the variants based on their effects of creating or disrupting the AAUAAA or AUUAAA signals. We found that 64 sites contain multiple AAUAAA or AUUAAA signals, and the associated genetic variants can have complicated effects on site strength. We showed the predicted values of these variants in Supplementary Data 5, but not Fig. S9.

### Identifying genetic variants regulating polyadenylation activity using deep/machine learning models

We applied the deep/machine learning models we developed to examine the genetic variants impacting polyadenylation activity. We analyzed the disease/trait-associated variants annotated by the ClinVar, UK BioBank, and GWAS Catalog,databases. To narrow down potentially impactful variants, we examined whether they affected the important cis-regulatory motifs we detected in Fig. 2. If a variant changed an important hexamer, we used a scanning window (step size: 1 nt) within +/−120 nt surrounding the variant, obtained the flanking 240 nt long sequence at each position, and performed the PolyaID and PolyStrength predictions. If the sequence showed PolyaID-predicted classification probability ≥0.9 for the wild-type or mutated sequence, we examined the variant's effect on polyA site usage, calculated using the PolyaStrength model. We showed the results centered around the maximum predicted cleavage site. One variant can be evaluated multiple times if it is associated with different genes encoded by the two opposite strands. If a site was located in the extended terminal exon region, we required that it be no more than 1 kb downstream RefSeq-

defined 3'-end, in order to limit this analysis to include only variants most likely to be functional. If a site was located in an intron, we used the logistic regression models we developed in Fig. 5 to predict the site expression probability. We required that the site should show expression probability ≥0.9 for either the wild-type or mutated sequence.

## Examine the APA regulation across human tissues/cells

We selected five different cell/tissue types to use to analyze tissue-specific APA regulation, including cerebellum, NSCs, ESCs, HTR8, and JEG3. We picked these cell types because they were generated by the same 3'READS protocol. We focused on analyzing the APA genes showing universal expression across the five cells/tissues. For each gene, we picked the top 2 expressed 3' terminal exonic sites supported by 3'READS to study. We examined the APA regulation using the Fisher Exact test comparing the proximal vs. distal site expression in each tissue vs. all other tissues. We found 3,297 genes showing APA regulation using the cutoff Benjamini-Hochberg adjusted $P$-values < 0.05, and an absolute usage change relative to the mean ≥0.1. Based on the unbiased principal component analyses of samples, we found cerebellum and NSCs (neuronal-like cells) were clustered together, while ESCs, HTR8, and JEG3 (embryonic-like cells) were grouped. We then identified proximal sites showing higher expression in neuronal-like cells, using the cutoff ≥10% relative usage difference in all neuronal-like cells vs. embryonic-like cells. Likewise, we also defined the sites showing higher usage across embryonic-like cells, and further divided them into two groups based on their relative higher usage levels (10–20% and >20%, respectively).

## Reporting summary

Further information on research design is available in the Nature Portfolio Reporting Summary linked to this article.

# Data availability

The data supporting the findings of this study are available from the corresponding authors upon request. The sequencing datasets analyzed in this study are available in the Gene Expression Omnibus (GEO) repository with the accession number GSE111134, GSE78657, GSE187694, GSE175163, and GSE219794.

# Code availability

The source codes were deposited in GitHub: https://github.com/zhejilab/PolyaModelsHuman and also from Zenodo[53].

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

## Acknowledgements

This work was supported by grants to Z.J.: the National Institutes of Health (R35GM138192, and R01HL161389), and the Lynn Sage Scholar fund. E.S. was supported by the Predoctoral Training Program in Biomedical Data Driven Discovery (T32LM012203). We thank Alfred George and the members of the Ji lab for helpful discussions.

## Author contributions

E.K.S. and Z.J. conceived and designed the study. E.K.S. performed the data analysis. E.K.S. and Z.J. wrote the manuscript. Z.J. supervised the research.

## Competing interests

The authors declare no competing interests.
