## [Peer Review File · Nature Communications]

Deep learning of human polyadenylation sites at nucleotide resolution reveals molecular determinants of site usage and relevance in diseaseREVIEWER COMMENTS

Reviewer #1 (Remarks to the Author):

Summary:

Stroup et al. develop two different types of human polyadenylation site predictors based on deep learning, one trained to classify polyadenylation sites from negative sequences (PolyaID) and another trained to regress the polyadenylation site affinity (PolyaStrength). In addition to classifying whether or not a sequence is a polyA site, PolyaID also predicts the corresponding cleavage distribution across the sequence. Both models are trained on a collection of publicly available 3'READS sequencing datasets from human tissues and cell types. The authors use the trained models to compute and visualize various importance metrics associated with motifs, e.g. positional influence of CFIm and CstF binding motifs. The authors further use the models to assess sequence determinants driving cleavage distribution heterogeneity, dissect genomic determinants influencing the choice of intronic sites and demonstrate the models' utility in scoring polyA signal-disrupting mutations.

Impact:

Overall I found this to be a well-executed and rigorous study with interesting insights into cis-regulatory polyadenylation regulation. The advances in model architecture, performance or biological insights are probably only incremental (as described in the paper, deep learning has by now been applied extensively to polyadenylation prediction). However, I do believe that this is one of the most well-designed polyadenylation signal predictors trained on endogenous transcriptomic data and provides an important orthogonal approach to models trained on MPRA data (such as APARENT 1/2 or PolyAPredictor).

I have some specific critiques in the comment section below, specifically regarding the lack of unbiased benchmarks and the lack of variant effect prediction benchmarks.

Strong points of the paper:

- Well-designed neural networks (including the losses used, etc.).
- Rigorous analyses of known cis-regulatory polyA motifs and their position-specific importance (Figure 2).
- Interesting cleavage heterogeneity analysis (Figure 3).
- Interesting intronic polyA site analysis (Figure 5).

Weak points:

- The variant prediction analysis in Figure 7 lacks any kind of benchmark comparison against measured effects.
- The model comparison in Figure 1 and S2 is only performed on the test set of the authors' processed data. Results may not be representative.

Comments and Suggested improvements:

- Major comment 1: The authors only benchmark their model to other published models on the test set of their own processed data (in Figure S2). It is thus unclear whether their model performance generalizes to other (well-established) datasets. I recommend performing additional benchmarks on relative usage measurements from PolyADB v3 (https://exon.apps.wistar.org/PolyA_DB/v3/) and/or APADB (<http://tools.genxpro.net:9000/apadb/>).

- Major comment 2: The authors do not perform any kind of benchmark of variant effect prediction in Figure 7, which I think they should in order to prove that their model is suitable as a variant interpretation tool. It would also be proper to compare to other polyadenylation predictors on this task. I suggest either using variant effect MPRA data from one of the following sources:

* Griesemer et al., 2021. <https://pubmed.ncbi.nlm.nih.gov/34534445/>

* Bogard et al., 2019. <https://pubmed.ncbi.nlm.nih.gov/31178116/>

* Vainberg Slutskin et al., 2019. <https://genome.cshlp.org/content/29/10/1635.full>

Or fine-mapped QTLs (where the task is to classify causal vs non-causal) from:

* Txrevise 3' events. https://www.ebi.ac.uk/eqtI/Data_access/

* Li et al., 2021. <https://pubmed.ncbi.nlm.nih.gov/33986536/>

Minor comments

- 7a / s7a: Do the Y-axes and colors both represent the same quantity (predictions), or something else? If they represent the same quantity I would suggest removing the color since it's confusing.

- s7a: I think GWAS Catalog only contains Tag SNPs, so this analysis doesn't say much since Tag SNPs are not necessarily estimated to be causal, they are just chosen to maximize LD for the genotyping. I suggest either removing this analysis or using fine-mapped variants, e.g. from UKBB (<https://www.finucanelab.org/data>).

- Be careful about the intronic/coding mutation examples in figure 7. Clearly the model has a high false positivity rate in predicting polyA activity in intronic sites (figure 4a), so it is not clear that these variants likely lead to a new (used) polyA isoform. I would rewrite the text to emphasize these intronic variants less.

Reviewer #2 (Remarks to the Author):

In this manuscript, Stroup and Ji developed new deep/machine learning models to identify polyA sites genome-widely. By using these models, they characterized the related cis-regulatory motifs and the

crosstalk of these cis-elements in determining polyA sites and cleavage heterogeneity. Furthermore, they investigated how local gene structure parameters determine the polyA site expression in introns and terminal exons. Finally, they applied their deep learning models to characterize disease/trait-associated genetic variants in altering polyadenylation activity. Overall, this is a very interesting paper that investigates the rules in regulating polyadenylation by deep/machine learning, and after an appropriate revision, the timely publication of this work at Nature Communications would advance the field and make a significant impact. However, some major questions and more data would need to be addressed to strengthen the manuscript.

- 1) What's the true performance of PolyalD and PolyaStrength models? Could the author plot it with scatter plot to show the correlation between modeling and true site usage level? In addition, AUROC is known to be strongly affected by imbalanced dataset. It's better to use AUPRC to evaluate the model performance. In addition, when comparing with the other publication deep learning polyA models, the same data sets would need to be used as a fair comparison and the related source code (jupyter notebook/R markdown) would need to be disclosed at github to allow reproducibility.
- 2) How the polyA site is predicted by the PolyalD and PolyaStrength? For example, in Fig. 1f, was the pre-mRNA sequence of HBA2 used for the prediction? Or, only the sequence surrounding the polyA sites was used? Some writing would be needed to clarify it.
- 3) To evaluate the scientific soundness, future readers would appreciate if all the computational analyses with raw source code for the plots and data analysis could be shared at GitHub. The current manuscript falls short in this part.
- 4) How many sites were used to characterize the cis-regulatory motifs? Why very few sites were plotted for the heatmap in Fig.2a?
- 5) It's very exciting that the authors utilized the deep learning model to reveal the crosstalk of cis-elements in polyA site definition. We noticed that the strength importance and the frequency had good correlation in most panels. In Fig.2c and Fig.2d, the motif frequency did not show much difference between upstream and downstream, while the importance score showed different pattern. Could the authors explain this discrepancy?
- 6) It's very interesting that the authors modeled in silico the mechanisms that determine the cleavage heterogeneity. To complete the work, could the authors validate these findings with mini-gene experiments?
- 7) In Fig. 4a, putative polyA sites were predicted by PolyalD in pre-mRNA. Does the exon-intron architecture influence the prediction of polyA sites in pre-mRNA?
- 8) What's the meaning of Fig. 4e? Is this a CDF plot for usage of different groups of polyA sites? The authors may need to clarify the plots in figure legends and methods.
- 9) We appreciate the efforts of authors to interpret the biological rules that determine the poly sites usage in Fig.4,5,6 through in silico modeling. The source code of regression models should be shared for readers. Moreover, Could the authors provide the related experimental validations to support these modeling findings? Some previous pioneer work from Bin Tian lab (e.g. PMID: 17210931) might be related to the new findings here.
- 10) Do the PolyalD and PolyaStrength have good correlation for in silico modeling, such as motif finding, identifying genetic variants regulating polyadenylation?
- 11) We appreciate the motif disruption/masking approach to quantify the importance of hexamers. The

saturation mutagenesis approach is also used to quantify the effect of cis-elements (e.g. PMID: 35581216). The latter may be a more direct way to show the position effect. Would the authors utilize this strategy to test how the sequence adjacent to polyA sites affect the polyA site determination?

Minor points:

- 1) For model comparison, were the same training and testing datasets used for different models?
- 2) For the box plots, the N number of each group should be clarified in figure legends

Reviewer #3 (Remarks to the Author):

In this study, the authors developed several deep-learning models that elucidate different facets of polyadenylation regulation and factors affecting the choice between alternative polyadenylation sites: (1) PolyAID is trained to, first, identify genomic sequences containing pA sites, and then predict, at nucleotide resolution, cleavage location; (2) PolyStrength predicts the strength/usage efficiency of pA sites. Using models derived from these tools, positional interactions between cis-regulatory pA motifs are defined and their optimal constellation is derived. In addition, logistic regression is used to identify gene-structure parameters that determine cleavage at intronic pA sites and affect usage of proximal 3'UTR pA sites. Last, the models are applied to examine the impact of disease-associated genetic variants on polyadenylation. Overall, this is a very nice study, the analyses are well conducted, and the text and figures are clear. The methodology and results make a significant advance in the field.

Major comments:

1. The results and models derived by this study are interesting and pose a significant advance over previous studies. But given that deep-learning methods have been already applied to the study of polyadenylation (as appropriately cited in the manuscript), novelty of the current work would be markedly elevated, if at least some effort is put in the attempt to delineate cell-type specific APA regulation. This aspect was not analyzed by previous DL studies in the field. An exhaustive analysis of this facet would merit a separate work, but some initial exploration of the capacity of DL methods to identify factors that determine cell-type specific pA regulation would significantly enhance the impact of the current study.
2. The last section of this study predicts the impact of disease-associated variants on polyadenylation. While I accept that experimental validations of selected candidates is out of the scope of this computational work, still some bioinformatics support for the predictions would markedly strengthen this analysis. For GWAS candidates, could the authors examine pA.QTL effects (using, for example, GTEx data – like was recently done in PMID: 32804959) and demonstrate that, in case detected, the effect is in the expected direction? Demonstrating colocalization of pA.QTL and eQTL (and GWAS) signals would be even stronger.

Minor comments:

1. The PolyStrength model was built using 53,105 pA sites from protein-coding genes. (a) were only 3'UTR pA sites used here or were intronic sites also included? (b) genes with a single pA site were excluded? (c) Was a minimal distance between the gene's pA sites applied?
2. Fig 2j presents an optimal configuration for pA sites. Are there genes conforming with this optimal set-up or is it selected against? If there are, do they have any common features (e.g., in terms of biological function? Expression level?)?
3. A cut-off of 10% change in PolyStrength score was used to call candidate variants. Does the proportion of (likely) pathogenic variants increase with this cut-off? If pA.OTL effects are examined as suggested above, are they associated with (likely) pathogenic annotation?
4. P16. The text says that 768 ClinVar variants were detected. Fig. S7c says N=756. Where this difference comes from?
5. P17. "For example, the genes LRPPRC, SCN4B, and RUNX1 are regulated by variants affecting the UGUA motif". This statement should be toned down. A regulatory effect is predicted, not demonstrated, by the current study. There are additional similar statements that should be toned down unless demonstrated experimentally.
6. Fig5c and 5i: the legend says: "we controlled for the polyA site strength and then compared ...". The figure compares features between 'Used' and 'Unused' sites. It is unclear which pA sites are shown – all sites or "controlled" ones? If all sites are shown, how "control" was applied in the results shown in these figs?
7. Figure S3f. I understood that this fig shows Delta with respect to AAUAAA. If so, why is there a bar for AAUAAA? (and if it is included, by definition, its value is 0)?

Point-by-point response

Reviewer #1 (Remarks to the Author):

Summary:

Stroup et al. develop two different types of human polyadenylation site predictors based on deep learning, one trained to classify polyadenylation sites from negative sequences (PolyaID) and another trained to regress the polyadenylation site affinity (PolyaStrength). In addition to classifying whether or not a sequence is a polyA site, PolyaID also predicts the corresponding cleavage distribution across the sequence. Both models are trained on a collection of publicly available 3'READS sequencing datasets from human tissues and cell types. The authors use the trained models to compute and visualize various importance metrics associated with motifs, e.g. positional influence of CFIm and CstF binding motifs. The authors further use the models to assess sequence determinants driving cleavage distribution heterogeneity, dissect genomic determinants influencing the choice of intronic sites and demonstrate the models' utility in scoring polyA signal-disrupting mutations.

Impact:

Overall I found this to be a well-executed and rigorous study with interesting insights into cis-regulatory polyadenylation regulation. The advances in model architecture, performance or biological insights are probably only incremental (as described in the paper, deep learning has by now been applied extensively to polyadenylation prediction). However, I do believe that this is one of the most well-designed polyadenylation signal predictors trained on endogenous transcriptomic data and provides an important orthogonal approach to models trained on MPRA data (such as APARENT 1/2 or PolyAPredictor).

I have some specific critiques in the comment section below, specifically regarding the lack of unbiased benchmarks and the lack of variant effect prediction benchmarks.

Strong points of the paper:

- Well-designed neural networks (including the losses used, etc.).*
- Rigorous analyses of known cis-regulatory polyA motifs and their position-specific importance (Figure 2).*
- Interesting cleavage heterogeneity analysis (Figure 3).*
- Interesting intronic polyA site analysis (Figure 5).*

Weak points:

- The variant prediction analysis in Figure 7 lacks any kind of benchmark comparison against measured effects.*
- The model comparison in Figure 1 and S2 is only performed on the test set of the authors' processed data.*

Results may not be representative.

We thank the reviewer for the detailed comments. And we performed all the suggested analyses which strengthened our conclusions.

Comments and Suggested improvements:

Major comments

- 1. The authors only benchmark their model to other published models on the test set of their own processed data (in Figure S2). It is thus unclear whether their model performance generalizes to other (well-established) datasets. I recommend performing additional benchmarks on relative usage*

measurements from PolyADB v3 (https://exon.apps.wistar.org/PolyA_DB/v3/) and/or APADB (<http://tools.genxpro.net:9000/apadb/>).

We agree that thorough and unbiased benchmarking is important. We now analyzed polyA sites defined by PolyA_DB v3.2 and APADB v2. We compared the performance of our models for the site identification (PolyaID vs. published) and site strength discrimination (PolyaStrength vs. published). Consistent with our previous results, our models outperformed published models. The results were added to Figures S2g-j and S3bc.

2. The authors do not perform any kind of benchmark of variant effect prediction in Figure 7, which I think they should in order to prove that their model is suitable as a variant interpretation tool. It would also be proper to compare to other polyadenylation predictors on this task.

I suggest either using variant effect MPRA data from one of the following sources:

- Griesemer et al., 2021. <https://pubmed.ncbi.nlm.nih.gov/34534445/>
- Bogard et al., 2019. <https://pubmed.ncbi.nlm.nih.gov/31178116/>
- Vainberg Slutskin et al., 2019. <https://genome.cshlp.org/content/29/10/1635.full>

Or fine-mapped QTLs (where the task is to classify causal vs non-causal) from:

- Txrevise 3' events. https://www.ebi.ac.uk/eqtl/Data_access/
- Li et al., 2021. <https://pubmed.ncbi.nlm.nih.gov/33986536/>

We thank the reviewer for the suggestion. We now made predictions for the MPRA data from Bogard et al. 2019, which was used for training the APARENT model. We compared our PolyaStrength vs. published models to distinguish highly vs lowly expressed sites from the MPRA library. The performance of PolyaStrength is nearly comparable to APARENT models and is better than PolyaPredictors (Figure S3d-f). As our model was trained based on endogenous polyA site expression across cells/tissues, its performance on MPRA data revealed its robustness.

Additionally, as suggested by the Reviewer 3, we analyzed a published cohort of PAS variants altering AAUAAA/AUUAAA and exhibited functional APA effects using GTEx RNA-seq data (PMID: 32804959). Our model can correctly capture polyA site strength changes upon the PAS mutations (Figure S9 and Table S5). The results proved that our model is suitable as a variant interpretation tool. And based on the analyses of these functional variants, we refined our cutoff to select significant variants.

Minor comments

1. Figure 7A / Supplemental Figure 7A: Do the Y-axes and colors both represent the same quantity (predictions), or something else? If they represent the same quantity I would suggest removing the color since it's confusing.

Our original Y-axes and the colors both represented the magnitude of prediction changes. A dot represented one polyA site. As suggested, we now removed the color to simplify the figures.

2. Supplemental Figure 7A: I think GWAS Catalog only contains Tag SNPs, so this analysis doesn't say much since Tag SNPs are not necessarily estimated to be causal, they are just chosen to maximize LD for the genotyping. I suggest either removing this analysis or using fine-mapped variants, e.g. from UKBB (<https://www.finucanelab.org/data>).

We thank the reviewer for pointing out the fine-mapped variants from UK BioBank. We now added the analyses of these variants and showed the results in Figure S11a-c and Table S6. We decided to keep the GWAS results as these may be useful for some researchers.

3. Be careful about the intronic/coding mutation examples in figure 7. Clearly the model has a high false positivity rate in predicting polyA activity in intronic sites (figure 4a), so it is not clear that these variants likely lead to a new (used) polyA isoform. I would rewrite the text to emphasize these intronic variants less.

The reviewer made a great suggestion. We now rewrote the genetics analyses section mostly emphasizing the variants located in 3'-most exons. And we toned down our discussion of intronic variants.

Reviewer #2 (Remarks to the Author):

In this manuscript, Stroup and Ji developed new deep/machine learning models to identify polyA sites genome-widely. By using these models, they characterized the related cis-regulatory motifs and the crosstalk of these cis-elements in determining polyA sites and cleavage heterogeneity. Furthermore, they investigated how local gene structure parameters determine the polyA site expression in introns and terminal exons. Finally, they applied their deep learning models to characterize disease/trait-associated genetic variants in altering polyadenylation activity. Overall, this is a very interesting paper that investigates the rules in regulating polyadenylation by deep/machine learning, and after an appropriate revision, the timely publication of this work at Nature Communications would advance the field and make a significant impact. However, some major questions and more data would need to be addressed to strengthen the manuscript.

We thank the reviewer for the supportive comments and detailed suggestions to improve our manuscript. And we performed the analyses to address all the reviewer's questions.

1. What's the true performance of PolyAID and PolyAStrength models? Could the author plot it with scatter plot to show the correlation between modeling and true site usage level? In addition, AUROC is known to be strongly affected by imbalanced dataset. It's better to use AUPRC to evaluate the model performance. In addition, when comparing with the other publication deep learning polyA models, the same data sets would need to be used as a fair comparison and the related source code (jupyter notebook/R markdown) would need to be disclosed at github to allow reproducibility.

When we compared the performance of our PolyAID and PolyAStrength to published models, we have always used the same datasets to evaluate. We now made it clear in the manuscript. As suggested, we added the AUPRC values in Figures S2 and S3. And as recommended by Reviewer 1, we also added benchmark results using polyA sites defined by PolyA_DB v3.2 and APADB v2. These results are consistent and support the better performance of our models.

To measure the performance of our PolyAStrength model, we evaluated its performance classifying the paired highly vs. lowly used polyA sites from individual genes. We did not compare the relative usage levels of polyA sites across different genes using the correlation. The reason is that multiple polyA sites from one gene can be stronger or weaker than all sites from another gene. It is not fair to compare polyA sites from different genes. Nevertheless, the Response Figure 1 shows the correlation between our PolyAStrength scores vs. the relative polyA site usage levels across genes ($R=0.52$). We now provided Jupyter notebooks containing the codes and raw data used to produce all figures (<https://github.com/zhejilab/PolyaModelsHuman>).

Response Figure 1. The correlation between the usage levels of terminal exonic polyA sites vs. PolyStrength scores. The Pearson correlation coefficient value is shown.

2. How the polyA site is predicted by the PolyAID and PolyStrength? For example, in Fig. 1f, was the pre-mRNA sequence of HBA2 used for the prediction? Or, only the sequence surrounding the polyA sites was used? Some writing would be needed to clarify it.

We used the pre-mRNA sequence as the input for the prediction. We showed example genomic regions (i.e. -100 to +100 region of HBA2 and GPATCH11 maximumly used polyA sites) in Figures 1f and S2m. We scanned the gene region using a 1-nt step size and made a prediction for each genomic position. For positions with a PolyAID classification >0.5 and center cleavage probabilities ≥ 0.05 , we averaged all overlapping cleavage vectors to create a cleavage profile. We now clarified these in the text and figure legends.

3. To evaluate the scientific soundness, future readers would appreciate if all the computational analyses with raw source code for the plots and data analysis could be shared at GitHub. The current manuscript falls short in this part.

We now provided Jupyter notebooks to produce all the figures and analyses (<https://github.com/zhejilab/PolyaModelsHuman>).

4. How many sites were used to characterize the cis-regulatory motifs? Why very few sites were plotted for the heatmap in Fig.2a?

Each row in the heatmap (Fig. 2a) showed the relative importance scores of a hexamer motif surrounding the polyA sites. It does not represent a polyA site. To measure the motif importance scores, we focused on 32,682 highly expressed polyA sites (≥ 100 3'READS & usage level $\geq 5\%$), as these sites are biologically more important than lowly expressed ones. For each hexamer at each position of the 240 nt surrounding these sites, we disrupted it and calculated the changes in PolyStrength scores to indicate the motif importance. The heatmap showed the sum important scores defined by $\text{sum}(\Delta\text{PolyStrength})$ across the polyA sites. This allowed us to obtain a position-specific motif importance profile surrounding the cleavage sites. The heatmap shows the aggregated motif importance relative to the cleavage site (columns) for 222 hexamers (rows) in 5 motif families that are recognized by different components of the polyadenylation machinery and were found to be important using our disruption approach. We now clarified this in the both the main text and figure legend.

5. It's very exciting that the authors utilized the deep learning model to reveal the crosstalk of cis-elements in polyA site definition. We noticed that the strength importance and the frequency had good correlation in most panels. In Fig.2c and Fig.2d, the motif frequency did not show much difference between upstream and downstream, while the importance score showed different pattern. Could the authors explain this discrepancy?

We agree with the reviewer's observation that generally the importance and motif frequency have a good correlation even in Fig. 2cd. As genomic polyA sites show quite variable motif compositions, different sets of polyA sites were used when we examined the crosstalk between paired motifs. This led to the differences between peak motif frequency and peak importance positions across panels.

6. It's very interesting that the authors modeled in silico the mechanisms that determine the cleavage heterogeneity. To complete the work, could the authors validate these findings with mini-gene experiments?

To address this question, we reanalyzed the published massively parallel reporter assay (MPRA) data from Bogard et al. 2019, where the studies introduced randomly mutated polyA site sequences to the cells and measured their relative expression and cleavage profiles using 3'-end sequencing. Using the MPRA data, we showed that polyA sites with multiple AAUAAAs show higher cleavage heterogeneity than those with only one AAUAAA, and sites with longer distance between the PAS AAUAAA and downstream U-rich or GU/CU-rich elements also show larger cleavage heterogeneity. We now added the analyses results to Figure S6j-l.

7. In Fig. 4a, putative polyA sites were predicted by PolyAID in pre-mRNA. Does the exon-intron architecture influence the prediction of polyA sites in pre-mRNA?

Our original analyses in the manuscript were based on the predictions from the nascent RNA sequences. The reason we did not consider spliced transcript is that the splicing complex bound to exon-exon junction may impact the polyadenylation machinery recruitment. But the reviewer raised an interesting question. We now made the prediction using spliced transcript sequences. We found that 17,806 novel polyA sites were predicted with the PolyAID classification probability > 0.5. These represent just 0.43% of total sites we predicted genome-wide and does not affect the conclusions we made. We agree that further investigation of the interplay between splicing and polyadenylation will be very interesting. But this requires a separate dedicated study to make solid conclusions. We now clarified in the manuscript that our predictions were based on the nascent RNA sequence.

8. What's the meaning of Fig. 4e? Is this a CDF plot for usage of different groups of polyA sites? The authors may need to clarify the plots in figure legends and methods.

Thanks for pointing this out. Fig. 4e is the CDF plot of polyA site usage levels. PolyA sites were grouped based on their genomic locations as in Fig. 4b. We only considered genes with ≥ 2 polyA sites in terminal exons. The relative usage of a polyA site was calculated as the ratio between the number of reads supporting the site and the summed reads from the top two expressed terminal exonic sites of the genes. We now clarified these in the figure legend.

9. We appreciate the efforts of authors to interpret the biological rules that determine the poly sites usage in Fig.4,5,6 through in silico modeling. The source code of regression models should be shared for readers. Moreover, Could the authors provide the related experimental validations to support these modeling findings? Some previous pioneer work from Bin Tian lab (e.g. PMID: 17210931) might be related to the new findings here.

We now provided Jupyter notebooks to produce the model and figures (<https://github.com/zhejilab/PolyaModelsHuman>). As the reviewer pointed out, previous published experimental data showed the competition between splicing and polyadenylation. Inhibiting of splicing activity by disrupting the splice sites or using anti-sense oligos to block U1 snRNP recruitment increased intronic polyadenylation (PMID: 17210931, PMID: 23874216, PMID: 20881964, PMID: 22770214). These experimental results validated some aspects of our analyses results. We now discussed these and cited the papers.

10. Do the PolyAID and PolyStrength have good correlation for *in silico* modeling, such as motif finding, identifying genetic variants regulating polyadenylation?

PolyAID and PolyStrength have high correlation for motif characterization, and variant effect predictions. In Figure S4b, we showed that the motif importance scores have a Pearson correlation of 0.996 between the PolyAID and PolyStrength models. When we predicted the effects of genetic variants, we required the site should have a PolyAID predicted probability >0.5 for either reference or alternative allele sequences and have the absolute PolyStrength score (log2) change >1. PolyStrength score has a higher dynamic range in differentiating highly vs. lowly used polyA sites than PolyAID, as we showed in the Figure 1. As suggested by reviewer 3, we analyzed the functional pA.QTL variants shown in the PMID: 32804959. The study revealed a few hundred genetic variants impacting PAS and altering APA activity analyzing RNA-seq data. In the Response Figure 2A, we selected the variants showing high PolyAID probability >0.9 for both reference and alternative allele sequences and plotted the histogram of their PolyStrength score changes (log2). We can see that PolyStrength score has a better dynamic range classifying the sites. In the Response Figure 2B, we showed the correlation between Δ PolyAID classification and Δ PolyStrength scores when predicting these pA.QTL variants. The two scores are correlated with the coefficient value 0.78. We showed the detailed Δ PolyAID and Δ PolyStrength scores in the supplementary tables for each variant examined (Table S5 and Table S6).

Response Figure 2. The changes of PolyAID and PolyStrength scores of pA.QTL variants, (A) For the sites with PolyAID probability >0.9 for both reference and alternative alleles, we plotted their PolyStrength score changes. (B) The correlation between the polyAID probability change vs. PolyStrength score change (alternative vs. reference). The Pearson correlation coefficient value is shown.

11. We appreciate the motif disruption/masking approach to quantify the importance of hexamers. The saturation mutagenesis approach is also used to quantify the effect of cis-elements (e.g. PMID: 35581216). The latter may be a more direct way to show the position effect. Would the authors utilize this strategy to test how the sequence adjacent to polyA sites affect the polyA site determination?

In Luo et al. (2022), motif importance is calculated by exhaustively mutating single nucleotides surrounding sites of interest and then running linear regressions to rank motif importance by slope and *P*-value. In this study, the hexamer disruption approach we used was based on the similar idea. We replaced a hexamer to 100 other random hexamers and calculated the change of model prediction values. We used the median change value to indicate the hexamer importance. The reason we used hexamer instead of single nucleotide is that hexamers represent functional motif units. We now tried the approach in PMID: 35581216. While we obtained expected results (positive correlation) for some hexamers such as AAUAAA and AUUAAA, we observed its limitation to examine motif importance of other less dominant PAS variants (such as AAAAAA, AGUAAA, and UAUAAA). We showed the example analysis results for these motifs at the -20 nt position of polyA sites in the Response Figure 3.

When we analyzed the PAS variants, because AAUAAA or AUUAAA show high occurrences and a dominant effect to promote polyA site formation in -20 nt of polyA sites, the single-nucleotide replacement of AAUAAA/AUUAAA into other PAS variants led to the decreased PolyAStrength scores and the negative correlation coefficient values. But these PAS variants are known to promote the polyA site formation. This example showed the limitation of the single-nucleotide replacement approach. The hexamer replacement approach is more appropriate to examine motifs contributing to polyA signals.

Response Figure 3. The single nucleotide saturation mutagenesis of PAS variants in -20 nt position upstream the cleavage sites. The +1 effect on motif indicates the motif gain effects (Δ PolyaStrength scores) upon single nucleotide mutations of all sequences. -1 represents the motif loss upon mutations, and 0 indicates the effects without the motif change.

Minor points

1. For model comparison, were the same training and testing datasets used for different models?

We have always used the same training and testing datasets for comparing the models. We now clarified this in the manuscript.

2. For the box plots, the *N* number of each group should be clarified in figure legends.

We thank the reviewer for pointing this out, we now added the *N* numbers in figure legends for the boxplots.

Reviewer #3 (Remarks to the Author):

In this study, the authors developed several deep-learning models that elucidate different facets of polyadenylation regulation and factors affecting the choice between alternative polyadenylation sites: (1) PolyAID is trained to, first, identify genomic sequences containing pA sites, and then predict, at nucleotide resolution, cleavage location; (2) PolyStrength predicts the strength/usage efficiency of pA sites. Using models derived from these tools, positional interactions between cis-regulatory pA motifs are defined and their optimal constellation is derived. In addition, logistic regression is used to identify gene-structure parameters that determine cleavage at intronic pA sites and affect usage of proximal 3'UTR pA sites. Last, the models are applied to examine the impact of disease-associated genetic variants on polyadenylation. Overall, this is a very nice study, the analyses are well conducted, and the text and figures are clear. The methodology and results make a significant advance in the field.

We thank the reviewer for the positive and constructive comments. And we performed all the suggested analyses which greatly improved our manuscript.

Major comments

1. The results and models derived by this study are interesting and pose a significant advance over previous studies. But given that deep-learning methods have been already applied to the study of polyadenylation (as appropriately cited in the manuscript), novelty of the current work would be markedly elevated, if at least some effort is put in the attempt to delineate cell-type specific APA regulation. This aspect was not analyzed by previous DL studies in the field. An exhaustive analysis of this facet would merit a separate work, but some initial exploration of the capacity of DL methods to identify factors that determine cell-type specific pA regulation would significantly enhance the impact of the current study.

We thank the reviewer for this suggestion. We now analyzed cell-type specific APA regulation using 3'READS data from 5 different tissues. These include neuronal stem cells (NSCs), cerebellum, embryonic stem cells (ESCs), and two placental cell lines HTR8 and JEG3. We selected these datasets because they were produced by the same 3'READS protocol. We compared the relative usage levels of proximal vs. distal polyA sites in 3'-terminal exons across the cell types (Figure S13). Using the principal component analyses, we observed NSCs and cerebellum (named as neuronal-like cells) show similar APA profiles, while ESCs, HTR8 and JEG3 (named as embryonic-like cells) were clustered together. In consistent with previous reports, more genes showed higher usage of proximal polyA sites in embryonic-like cells than neuronal-like cells (1,474 vs. 266 genes, based on the cutoff Benjamini-Hochberg corrected P -value < 0.05 & usage level difference $> 10\%$). Proximal polyA sites showing higher usage in embryonic-like cells tend to have lower PolyStrength scores and are located far away from distal sites, compared to those showing higher usage in neuronal-like cells. These results showed that our deep learning models can be useful for the future investigation of cell-type specific APA regulation. We now added the analyses in the discussion section and Figure S13.

2. The last section of this study predicts the impact of disease-associated variants on polyadenylation. While I accept that experimental validations of selected candidates is out of the scope of this computational work, still some bioinformatics support for the predictions would markedly strengthen this analysis. For GWAS candidates, could the authors examine pA.QTL effects (using, for example, GTEx data – like was recently done in PMID: 32804959) and demonstrate that, in case detected, the effect is in the expected direction? Demonstrating colocalization of pA.QTL and eQTL (and GWAS) signals would be even stronger.

We thank the reviewer for pointing out this very useful benchmark dataset and analyses. In Shulman and Elkon 2020, the authors identified a few hundred functional PAS variants associated with site usage changes measured by RNA-seq. We now perform PolyAID and PolyStrength prediction of the reference

and alternative allele sequences. As shown in Figure S9, we grouped the sites based on their mutated PAS types, and our models can capture the regulation of the PAS loss or gain in the expected direction. And we now provided a new supplementary table (Table S5) to show the detailed prediction results.

Importantly, this analysis refined our cutoff to identify functional variants altering polyadenylation activity. We examined the distribution of PolyAID classification probabilities, PolyStrength scores, and Δ PolyStrength values of the pA.QTL variants. Our original cutoff to select variants (>10% change of absolute PolyStrength values) is too stringent. Based on the analyses, our new cutoff to select potential functional variants is PolyAID classification probability ≥ 0.5 and PolyStrength ≥ -9 for either reference or alternative allele sequence, and the variant induce a change of \log_2 PolyStrength scores ≥ 1 . This new cutoff conservatively identified 86.7% of PAS pA-QTLs as significant variants. We now used this new cutoff and refined the genetic variants analyses results in Figures 7, S9, S11 and S12.

Minor comments

1. The PolyStrength model was built using 53,105 pA sites from protein-coding genes. (a) were only 3'UTR pA sites used here or were intronic sites also included? (b) genes with a single pA site were excluded? (c) Was a minimal distance between the gene's pA sites applied?

When we trained the PolyStrength model, we used terminal exonic sites from protein-coding genes. We did not use the intronic sites. And we required that the genes should contain ≥ 2 polyA sites in terminal exons. The single pA sites were not included in the analyses. No minimum distance between each gene's polyA sites was imposed. We now clarified this in the Methods section.

2. Fig 2j presents an optimal configuration for pA sites. Are there genes conforming with this optimal set-up or is it selected against? If there are, do they have any common features (e.g., in terms of biological function? Expression level?)?

We thank the reviewer for raising this very interesting analyses. We now analyzed 13,876 highly expressed polyA sites (≥ 100 PASS reads and usage level $\geq 5\%$) with the PAS AAUAAA (CPSF binding) and at least one downstream U-rich or GU-rich element (CstF binding). The motif compositions of these polyA sites tend to be diverse (Fig. S5a-c). Besides the CPSF and one CstF motif, 44.9% of these sites contained one additional optimally positioned motif, 27.4% of sites contain two, 2.5% of sites are with three, and very few sites (0.14%) have all other four motifs (Fig. S5a-c). Sites with more optimized motifs are generally stronger and are likely to be the single or distal sites of the genes (Fig. S5d-e). But genes showing different polyA site motif compositions are expressed at comparable levels (Fig. S5f). Indeed, besides the polyadenylation activity, the gene expression levels are also determined by other regulatory layers such as transcription initiation. The gene ontology analyses of genes with ≥ 5 optimized motifs showed that they are enriched in the pathways such as "positive regulation of macromolecule metabolic process", "cellular response to stress", and "positive regulation of transcription, DNA-templated" (Fig. S5g). Altogether, the results showed that genomic polyA sites tend to have diverse motif configurations which potentially allow differential regulation of site usage during biological processes. We added these results to the manuscript.

3. A cut-off of 10% change in PolyStrength score was used to call candidate variants. Does the proportion of (likely) pathogenic variants increase with this cut-off? If pA.QTL effects are examined as suggested above, are they associated with (likely) pathogenic annotation?

We now changed the cutoff to select candidate functional variants based on the pA.QTL effects described above. As suggested, we grouped the variants based on their changes in the $|\Delta$ PolyStrength| and plot

the distribution of associated clinical significance for each group. As shown in Response Figure 4, we did not observe significant changes of pathogenic annotations of the variants using different cutoffs.

Response Figure 4. We selected significant ClinVar variants altering polyadenylation activity using different $|\Delta\text{PolyaStrength}|$ cutoffs and then examined their annotated risks.

4. P16. The text says that 768 ClinVar variants were detected. Fig. S7c says N=756. Where this difference comes from?

There were some variants which can impact genes located in two different strands. And these variants were evaluated multiple times, once for each gene. To avoid the confusion, we now changed the numbers to variant-pA combinations in both text and figures. And we noted in the method that one variant can be evaluated multiple times if it is associated with different genes encoded in two opposite strands.

5. P17. “For example, the genes *LRPPRC*, *SCN4B*, and *RUNX1* are regulated by variants affecting the UGUA motif”. This statement should be toned down. A regulatory effect is predicted, not demonstrated, by the current study. There are additional similar statements that should be toned down unless demonstrated experimentally.

We now rewrote the sentences saying that the variants were “predicted” to have a regulatory effect.

6. Fig5c and 5i: the legend says: “we controlled for the polyA site strength and then compared ...”. The figure compares features between ‘Used’ and ‘Unused’ sites. It is unclear which pA sites are shown – all sites or “controlled” ones? If all sites are shown, how “control” was applied in the results shown in these figs?

We originally put the plot showing the controlled sites in the supplementary figure. As the reviewer pointed out, this critical plot is important for readers to understand the analyses. We now moved the panels showing the controlled site strengths to the main figure (Fig. 5c and 5i).

7. Figure S3f. I understood that this fig shows Delta with respect to AAUAAA. If so, why is there a bar for AAUAAA? (and if it is included, by definition, its value is 0)?

In our original plot, we included the bar for AAUAAA and the value was 0. As the reviewer pointed out, this may cause confusion and we now removed the AAUAAA bar.

REVIEWERS' COMMENTS

Reviewer #1 (Remarks to the Author):

The authors sufficiently addressed all of my previous comments.

I have two new minor comments:

1. For the new UK Biobank analysis, in Figure S11a the caption starts off with "Effects of fine-mapped variants from the UK BioBank ...". It is not described where the authors obtained these fine-mapped variants from (cite the source), or if the authors performed the fine-mapping themselves (in that case mention this in Methods).

2. It would be great if the authors could deposit their processed version of the training data online (the PAS sequences with measured cleavage count profiles; i.e. what the model is trained on; only total counts are currently shown in Supp Table 2), and mention the data location in the data availability statement.

If the authors address these two minor details (the editor is free to assess this without my input), then I recommend accepting the paper for publication.

Reviewer #2 (Remarks to the Author):

The authors have comprehensively addressed our review comments and their manuscript has been improved accordingly. We would recommend it being accepted and get published at Nature Communications soon, and this work will have an important impact to the scientific community.

Reviewer #3 (Remarks to the Author):

The authors have addressed all my comments.

I have one last minor comment which refers to the analysis of Clinvar variants.

The text says (page 18): "57% of disease-associated ClinVar variants were annotated as "uncertain significance" and their regulatory impacts on polyadenylation activity shown here could be the molecular mechanism mediating their association with diseases/traits (Fig. 7c)".

Clinvar variants annotated as 'uncertain significance' (let alone those annotated as 'benign / likely benign') are NOT "disease-associated" variants and should not be referred to as such. These variants are located within disease-associated genes, but they are not likely to have any pathogenic effect. The way the current text is phrased suggests that all these variants are (likely) pathogenic, but that their

pathogenic mechanism is unknown at present. This sentence might be misleading and should be rephrased.

Point-by-point response

Reviewer #1 (Remarks to the Author):

The authors sufficiently addressed all of my previous comments.

We thank the reviewer for taking the time reviewing our manuscript.

I have two new minor comments:

1. For the new UK Biobank analysis, in Figure S11a the caption starts off with "Effects of fine-mapped variants from the UK BioBank ...". It is not described where the authors obtained these fine-mapped variants from (cite the source), or if the authors performed the fine-mapping themselves (in that case mention this in Methods).

As suggested, we now added the reference.

2. It would be great if the authors could deposit their processed version of the training data online (the PAS sequences with measured cleavage count profiles; i.e. what the model is trained on; only total counts are currently shown in Supp Table 2), and mention the data location in the data availability statement.

If the authors address these two minor details (the editor is free to assess this without my input), then I recommend accepting the paper for publication.

As suggested, we now added the number of reads between -50 and +50 nt around the representative cleavage sites in DataS2, which we used to calculate the cleavage profiles. Additionally, we put our processed version of the training data including the PAS sequences with measured cleavage count profiles on GitHub. The file name is "PolyaID_training_data.txt".

Reviewer #2 (Remarks to the Author):

The authors have comprehensively addressed our review comments and their manuscript has been improved accordingly. We would recommend it being accepted and get published at Nature Communications soon, and this work will have an important impact to the scientific community.

We thank the reviewer for taking the time reviewing our manuscript and the encouraging comment.

Reviewer #3 (Remarks to the Author):

The authors have addressed all my comments.

I have one last minor comment which refers to the analysis of Clinvar variants.

The text says (page 18): "57% of disease-associated ClinVar variants were annotated as "uncertain significance" and their regulatory impacts on polyadenylation activity shown here could be the molecular mechanism mediating their association with diseases/traits (Fig. 7c)". Clinvar variants annotated as 'uncertain significance' (let alone those annotated as 'benign / likely benign') are NOT "disease-associated" variants and should not be referred to as such. These variants are located within disease-associated genes, but they are not likely to have any

pathogenic effect. The way the current text is phrased suggests that all these variants are (likely) pathogenic, but that their pathogenic mechanism is unknown at present. This sentence might be misleading and should be rephrased.

We thank the reviewer for taking the time reviewing our manuscript. As suggested, we deleted the word "disease-associated".